# Rep15 interacts with several Rab GTPases and has a distinct fold for a Rab effector

Amrita Rai [1✉], Anurag K. Singh [2], Nathalie Bleimling[1], Guido Posern[2], Ingrid R. Vetter[3] & Roger S. Goody [1✉]

In their GTP-bound (active) form, Rab proteins interact with effector proteins that control downstream signaling. One such Rab15 effector is Rep15, which is known to have a role in receptor recycling from the endocytic recycling compartment but otherwise remains poorly characterized. Here, we report the characterization of the Rep15:Rab15 interaction and identification of Rab3 paralogs and Rab34 as Rep15 interacting partners from a yeast two-hybrid assay. Biochemical validation of the interactions is presented and crystal structures of the Rep15:Rab3B and Rep15:Rab3C complexes provide additional mechanistic insight. We find that Rep15 adopts a globular structure that is distinct from other reported Rab15, Rab3 and Rab34 effectors. Structure-based mutagenesis experiments explain the Rep15:Rab interaction specificity. Rep15 depletion in U138MG glioblastoma cells impairs cell proliferation, cell migration and receptor recycling, underscoring the need for further clarification of the role of Rep15 in cancer.

[1] Department of Structural Biochemistry, Max Planck Institute of Molecular Physiology, Dortmund, Germany. [2] Institute for Physiological Chemistry, Martin Luther University Halle-Wittenberg, Halle (Saale), Germany. [3] Department of Mechanistic Cell Biology, Max Planck Institute of Molecular Physiology, Dortmund, Germany. ✉email: amrita.rai@mpi-dortmund.mpg.de; goody@mpi-dortmund.mpg.de

Vesicular trafficking involves a precisely controlled flow of transport vesicles between the endocytic and secretory/exocytic pathways and this flow is tightly controlled by Rab GTPases[1]. Rab GTPases have defined membrane locations and mediate the exchange of cargos between different organelles via selective binding and recruiting of distinct Rab interacting proteins[2]. Rab GTPases are the largest branch of the small GTPase family and more than 70 Rab family members have been reported so far. Similar to other small GTPases, Rab GTPases act as molecular switches. In the GTP-bound form, Rab is in an active state (on) and GDP-bound Rab is inactive (off)[1]. Two regulatory proteins tightly control the cycling between the on and off states. These are guanine nucleotide exchange factors (GEFs) that catalyze exchange of GDP by GTP and GTPase activating proteins (GAPs) that deactivate Rab by increasing the rate of GTP hydrolysis[3–5]. GTP-bound Rabs interact with various downstream effector molecules and Rab:effector complexes spatiotemporally control the distinct steps of vesicular trafficking[1]. Rab:effector recognition provides functional specificity and determines the various stages of trafficking. Most Rabs can interact with more than one effector molecule and similarly, effectors are also capable of interacting with phylogenetically similar Rabs[6–8]. Dysfunction of Rabs or their regulators or effector proteins contributes to various disease states[9–11].

Rab15 is enriched explicitly in the brain and its distinct isoforms are expressed in neuroblastoma cells[12,13]. In CHO cells, Rab15 co-localizes with Rab 4 and 5 on early endosomes/sorting endosomes (SE) and also with Rab11 on the endocytic recycling compartment (ERC)[14], suggesting that it has distinct roles in the different endocytic compartments which can be achieved by recruiting different sets of effector molecules. In previous work, we have shown that Rab15 interacts with a plethora of bMERB family member effectors such as MICAL1, MICAL3, MICAL-cL, EHBP1 and EHBP1L1[8,15,16]. However, additional studies are required to elucidate the functional significance of the Rab15:bMERB interaction. Apart from bMERB family members, Rep15 is another Rab15 effector molecule that regulates transferrin receptor trafficking. Recently multi-omics analysis data suggested REP15 as a novel colorectal cancer-specific driving gene[17]. Rep15 interacts with Rab15 at the ERC, and overexpression and depletion of Rep15 by si RNA attenuates the recycling of internalized transferrin receptor recycling via ERC without affecting receptor trafficking through sorting endosomes[18]. Rep15 is one of the least well-characterized Rab effectors; hence there remains a large gap in understanding the binding mode of the Rep15:Rab15 interaction and it also raises the question as to whether Rep15 only interacts with Rab15 or also has other uncharacterized interaction partners.

Rab3 family members are also brain and neuro-endocrinal enriched GTPases and are implicated in the secretory pathway[19]. The mammalian Rab 3 subfamily is comprised of 4 paralogs (Rab3A, 3B, 3 C, and 3D). Rab3 proteins are associated with secretory vesicles and have a role in the secretion of hormones and neurotransmitters[20]. Active Rab3 is localized to secretory vesicles and synaptic vesicles and dissociates upon depolarization of the nerve terminal. Active Rab3 has been shown to interact with various effector proteins, including Rabphilin3A[21], Rabin3A[22], RIM[23], Noc2[24], synapsin[25], Myo5a[26], Slp4-a and NMHC IIA[27]. Single/double knockout mice of Rab3 paralogs are viable, whereas a triple knockout was lethal only if Rab3A was deleted. Knockout mice lacking all four mammalian RAB3 paralogs (RAB3A–D) show perinatal lethality, however, they show no changes in synaptic organization and only a minor change in neurotransmitter release[28]. This might be due to the presence of highly expressed Rab27A/B on the synaptic vesicles, since Rab3 and Rab27 share some common effector molecules[29].

Besides having a role in vesicular transport and early development, Rab 3 family members are also implicated in several cancers[30], but effector molecules involved are still to be identified.

Here, we describe the mechanistic basis of Rep15:Rab interactions and show that Rep15 forms a stable complex with Rab15 with 1:1 stoichiometry. Using the yeast two-hybrid approach, we have identified Rab3 paralogs and Rab34 as novel Rep15 interaction partners and further validated the interaction by co-localization and biochemical experiments. To decipher the binding mode, we have determined the structure of Rep15:Rab3B and Rep15:Rab3C complexes, respectively. The structures show that Rep15 is a globular Rab effector. Detailed biochemical studies reveal that Rep15 is selective for Rab15, Rab3 paralogs and Rab34 and does not bind to other evolutionarily similar Rab GTPases. Mutational analyses of both Rep15 and Rab3A reveal the molecular basis for Rab selectivity. Rep15 knockout experiments in U138MG glioblastoma cells show that Rep15 depletion leads to a decrease in cell proliferation and cell migration and also influences transferrin recycling.

## Results and discussion

**Biochemical characterization of Rep15:Rab15 interaction.** Rep15 is composed of 236 amino acids (UniProt ID: Q6BDI9) and in the present work, a naturally occurring N101D Rep15 variant (VAR_039548) was used (Supplementary Fig. 1a). SMART and IterPro database searches show that the Rep15 has no putative functional domains and shares no homology with any known Rab effector or other proteins. By yeast-two hybrid and pull-down approaches, Strick et al., have shown that the Rep15 is an effector molecule for Rab15 and co-localizes to endosomal membranes[18]. However, the biochemical characterization of this complex is not available. To understand the mechanism underlying Rep15:Rab15 complex formation, we purified full-length Rep15 and Rab15$_{GppNHp}$ and monitored their complex formation by analytical size exclusion chromatography (aSEC). Further, the interaction affinity was quantified by isothermal titration calorimetry (ITC) measurements (Fig. 1a, b). As expected, the formation of a stable complex of Rep15:Rab15 with a $K_D$ value of $0.47 \pm 0.12 \, \mu M$ was detected and the binding energy has a similar contribution from both enthalpic and entropic components.

In parallel, association ($k_{on}$) and dissociation ($k_{off}$) rate constants were determined by transient kinetic experiments using a stopped-flow apparatus. For association rate constant measurements, $1 \, \mu M$ Rab15 loaded with fluorescent mant-GppNHp (2′/3′-O-(N-methyl-anthraniloyl)-guanosine-5'-[(β,γ)-imido]triphosphate) was mixed rapidly with increasing concentrations of Rep15. The association of mant-GppNHp Rab15 with the Rep15 leads to an increase in fluorescence polarization and the observed pseudo-first-order rate constant values ($k_{obs}$) values were plotted against increasing concentration of Rep15. The association rate constant for Rep15 is $8.16 \times 10^6 \, M^{-1} \, s^{-1}$ and the intercept on the y-axis yielded the dissociation rate constant of $4.73 \, s^{-1}$ (Fig. 1c). The equilibrium dissociation constant ($K_D$) for the complex was calculated as the ratio of $k_{off}/k_{on}$ to have a value of $0.58 \, \mu M$, in reasonable agreement with the value of $0.47 \pm 0.12 \, \mu M$ obtained in ITC experiments. To measure the dissociation rate constant directly, the Rep15:mant-GppNHp Rab15 complex was rapidly mixed with a 25-fold excess of GppNHp Rab15. Dissociation of mant-GppNHp Rab15 was biphasic, with a faster dissociation rate constant of $2.28 \, s^{-1}$ ($k_{off1}$) and a slower dissociation rate constant of $0.47 \, s^{-1}$ ($k_{off2}$) was observed (Fig. 1d). The faster dissociation rate constant was in line with the rate constant measured indirectly from the plotting association rate constants against the Rep15 concentration (Fig. 1c). To rule out the possibility that the two dissociation rate constants are due to the usage of mant-

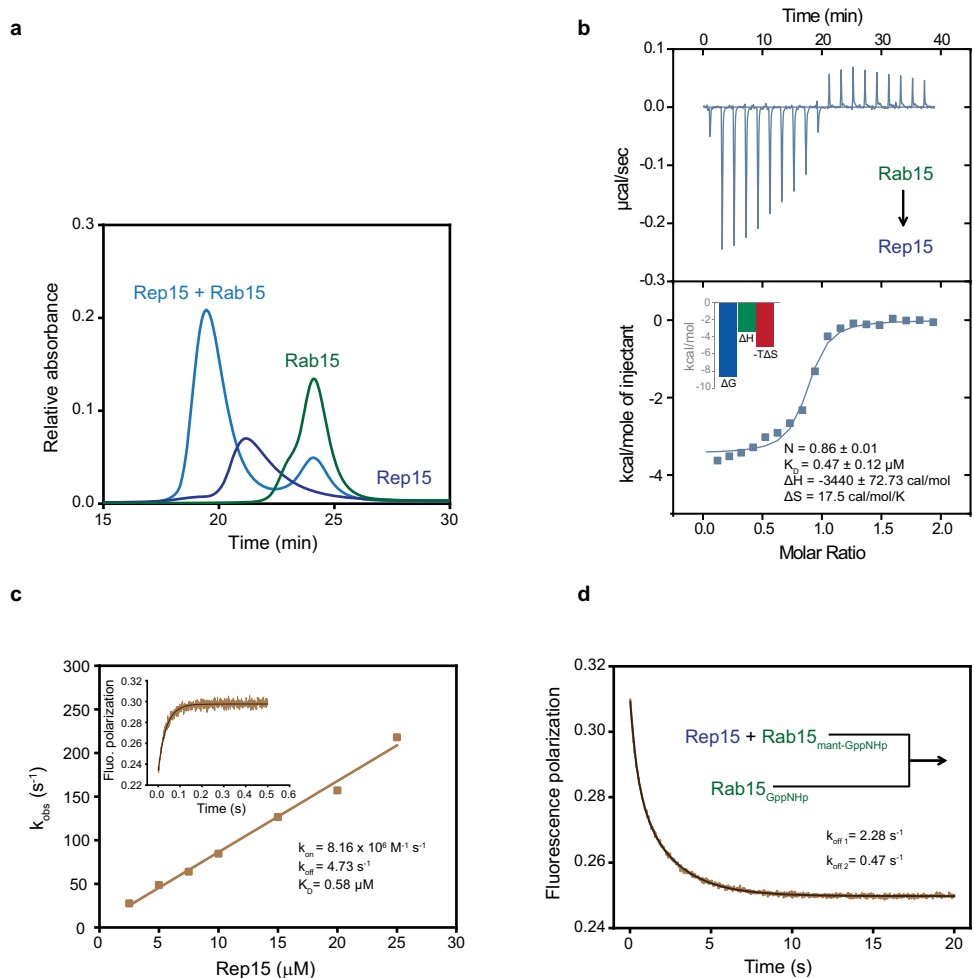

**Fig. 1 Biochemical characterization of Rep15:Rab15 interaction. a** Interaction of Rep15 with Rab15$_{GppNHp}$ was analyzed by analytical size exclusion chromatography. 110 μM of Rep15 (blue), 121 μM of Rab15$_{GppNHp}$ (green) and the mixture of both (sky blue) is loaded on a Superdex 75 10/300 GL column. A clear complex formation was observed. The curves shown are representative of at least three experiments. **b** ITC measurement of Rep15 (50 μM) titrated with 500 μM of Rab15$_{GppNHp}$. Heat peaks were integrated and fitted to a one-site-binding model yielding the binding stoichiometry (N), the enthalpy (ΔH), the entropy (ΔS) and the dissociation constant (K$_D$). The data are representative of at least three repetitions. **c** Association rate constant ($k_{on}$) for the Rab15$_{mant-GppNHp}$ (0.5 μM) interaction with the different concentration of Rep15 (2.5–25 μM). Association was monitored by the change in fluorescence polarization using a stopped-flow apparatus at 25 °C. Association of the Rep15 with Rab15$_{mant-GppNHp}$ leads to an increase in fluorescence polarization and data is fitted to a single exponential. The change in fluorescence polarization of 1 μM Rab15$_{mant-GppNHp}$ upon 15 μM Rep15 binding is shown in the inset. **d** Dissociation of Rab15 from the Rep15 was determined by monitoring the decrease of fluorescence polarization after mixing a complex of Rep15:Rab15$_{mant-GppNHp}$ (2 μM) with a 25-fold excess of nonfluorescent Rab15$_{GppNHp}$ and data is fitted to a double exponential. Data analyzed are averages of 6–8 individual traces.

GppNHp, we repeated the experiment with (3'-O-(N-Methyl-anthraniloyl)−2'-deoxyguanosine-5'-[(β,γ)-imido]triphosphate) mant-dGppNHp loaded Rab15. However, the biphasic dissociation persists (Supplementary Fig. 1c, d), suggesting the possibility of a two-step mechanism.

To help characterize the exact Rab15 binding region/domain, we generated a series of Rep15 deletion constructs based on the secondary structure prediction (Supplementary Fig. 1a). The first construct lacks the N-terminal 16 amino acids (Rep15 ΔN1); the second one contains residues 17 to 115 (Rep15 ΔNC1). In contrast, the construct REP15 ΔN2 lacks the first 139 amino acids and the last construct is composed of amino acids 17-202 (Rep15 ΔNC2). We expressed MBP-fusion constructs and could only successfully purify REP15 ΔN1, while the other three constructs are not stable after MBP tag removal by TEV protease. REP15 ΔN1 forms a stable complex with Rab15 (Supplementary Fig. 1b). Until now, no structural information is available for any Rep15 orthologs. Therefore, we performed a crystallization screen with

Rep15 and Rep15 ΔN1. However, we failed to obtain crystals. To reveal the binding mode, we attempted the crystallizion of full-length Rep15, as well as Rep15 ΔN1, with GppNHp loaded Rab15 by mixing the purified proteins in a 1:1 ratio, but all trials were unsuccessful. Next, we co-expressed and purified the Rep15 ΔN1:Rab15$_{Q67L\_1-176}$ (active) complex and set up the crystallization screen; we also performed surface lysine methylation of the Rep15:Rab15 complex and also limited proteolysis of Rep15:Rab15$_{GppNHp}$ complex during crystallization; however, none of these trials yielded crystals.

**Rab3 paralogs and Rab34 as novel Rep15 interacting partners**. Since it has been shown that a number of different effector molecules can bind to evolutionarily similar Rabs, and Rep15 had not been characterized in detail, to gain insight into the function of Rep15, we set out to identify new Rep15 interaction partners. Thus, we performed a yeast two-hybrid (Y2H) screen using

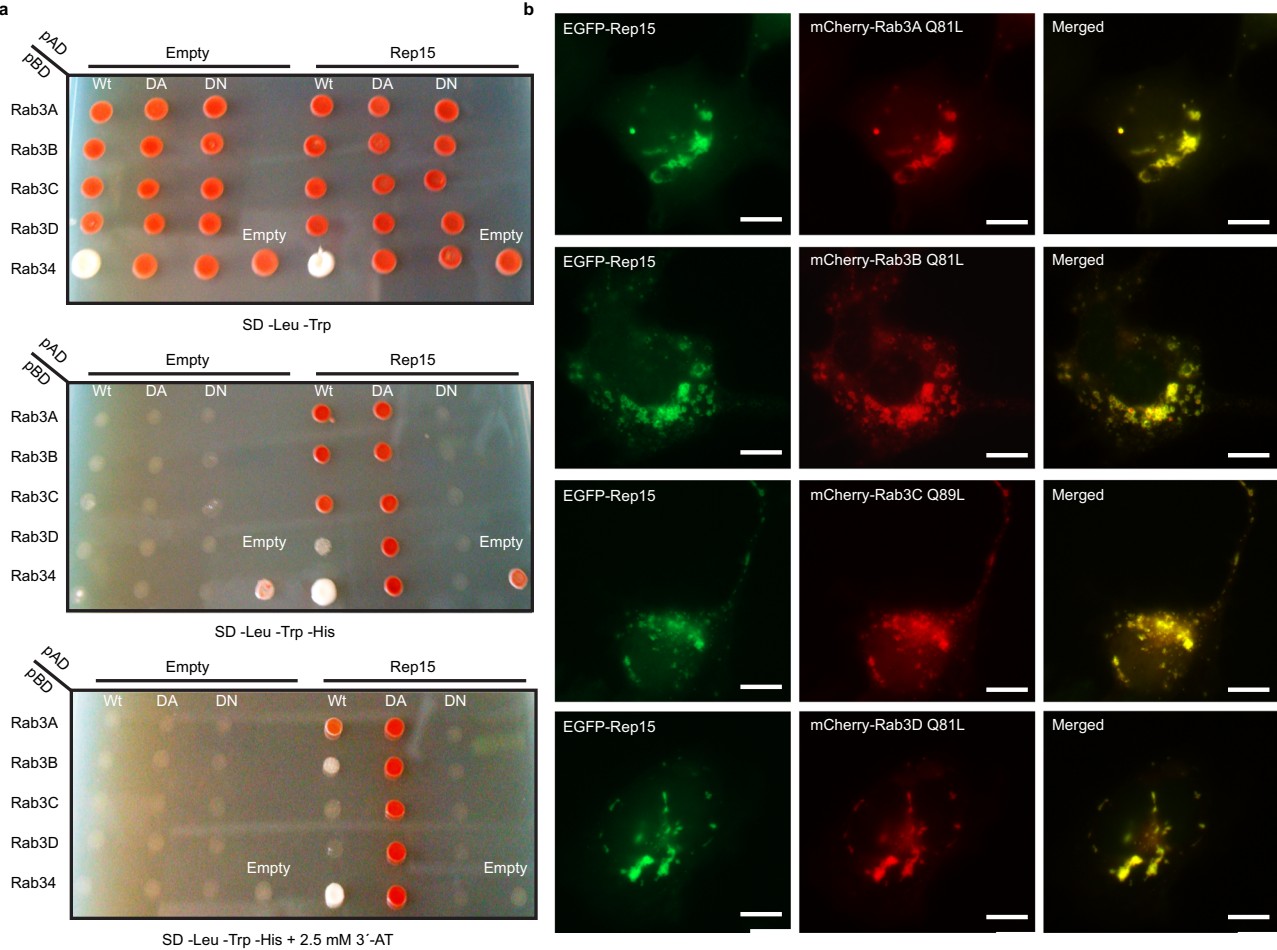

**Fig. 2 Identification and validation of novel Rep15 interaction partners. a** A yeast two-hybrid screen between Rep15 (pGADT7) and wild type (Wt), dominant active (QL) and dominant inactive (SN/TN) form of Rab3A, Rab3B, Rab3C, Rab3D and Rab34 (pLexA) was performed. The cells were grown on SD-LW, SD-LWH and SD-LWH plus 2.5 mM 3-Aminotriazol (3′-AT), a competitive inhibitor of the HIS3 gene product. Y2H analysis identified interactions between the Rep15 and Rab3 paralogs/Rab34. **b–e** Immunofluorescence of transiently co-expressed EGFP-Rep15 and mCherry-Rab3 paralogs in Cos7 cell line by fluorescence microscopy. Merged images show strong co-localization between EGFP-Rep15 and mCherry tagged active Rab3 paralogs. Scale bars: 10 μm. Experiments were repeated at least three times independently with similar results.

Rep15 against a constitutively active (dominant active; GTP bound) human Rab library. We detected Rab3 family members and Rab34 as Rep15 interacting partners. However, we could not detect the Rep15:Rab15 interaction in our Y2H assay (Supplementary Fig. 2), suggesting the possibility of missing other potential Rabs as Rep15 interaction partners. To rule this out, additional experiments are needed, for example, pull-downs against different human Rabs. To validate the results, we repeated Y2H experiments with Rep15 against wild-type (wt), constitutively active, and constitutively inactive (dominant-negative; GDP bound) forms of the Rab3 paralogs and Rab34. As expected, we observed interaction in the case of dominant active Rab constructs and also with wild type Rab GTPase in the case of Rab3A, Rab3B and Rab34. No interaction was observed in the case of dominant-negative Rab constructs, indicating that Rep15 is an effector for Rab3 paralogs and Rab34 (Fig. 2a). Similarly to Rep15, Rab3 paralogs are also highly expressed in the gastrointestinal tract, endocrine tissue[31], and regulate synaptic vesicular trafficking. Rab34 is a Golgi-resident RabGTPase[32] and active Rab34 shifts to the lysosome, where it regulates lysosomal morphology together with its effector RILP[33], and also has a role in phagosome-lysosomal fusion[34].

For Rab:effector complexes to be functional, their spatiotemporal localization is crucial. To evaluate whether Rep15 shares the same membrane identity as that of Rab3 paralogs, we transiently overexpressed full-length Rep15 and Rab3 paralogs in Cos7 cells and could show that Rep15 co-localizes with all active Rab3 paralogs (Fig. 2b–e), indicating that apart from having a role in recycling endosomes, Rep15 might have roles in the Rab3/Rab34 dependent process.

**Biochemical characterization of Rep15:Rab3 and Rep15:Rab34 interaction**. For biochemical characterization, we purified the full-length Rep15 and Rabs from *E. coli* expression systems and preparatively loaded Rabs with GppNHp. However, apart from GppNHp, Rab34 is also bound to $GppNH_2$ (hydrolysis product of GppNHp). To overcome the problem of heterogeneity, we expressed the dominant active $Rab34_{Q111L\_1-237aa}$ that has reduced GTPase activity and purified the protein in the presence of GTP. Next, we monitored complex formation via aSEC (analytical size exclusion chromatography) experiments and stable complex formation was detected with Rab3 paralogs and Rab34 (Fig. 3a–d, i). To quantitatively characterize the interaction, we carried out ITC experiments and observed $K_D$ values in the range of 0.48–2.16 μM for Rab3 paralogs in an enthalpy-driven process (Fig. 3e–h). The affinities are similar to that observed in the case of Rab15, whereas the highest affinity of ~0.1 μM ($64 \pm 32$ nM)

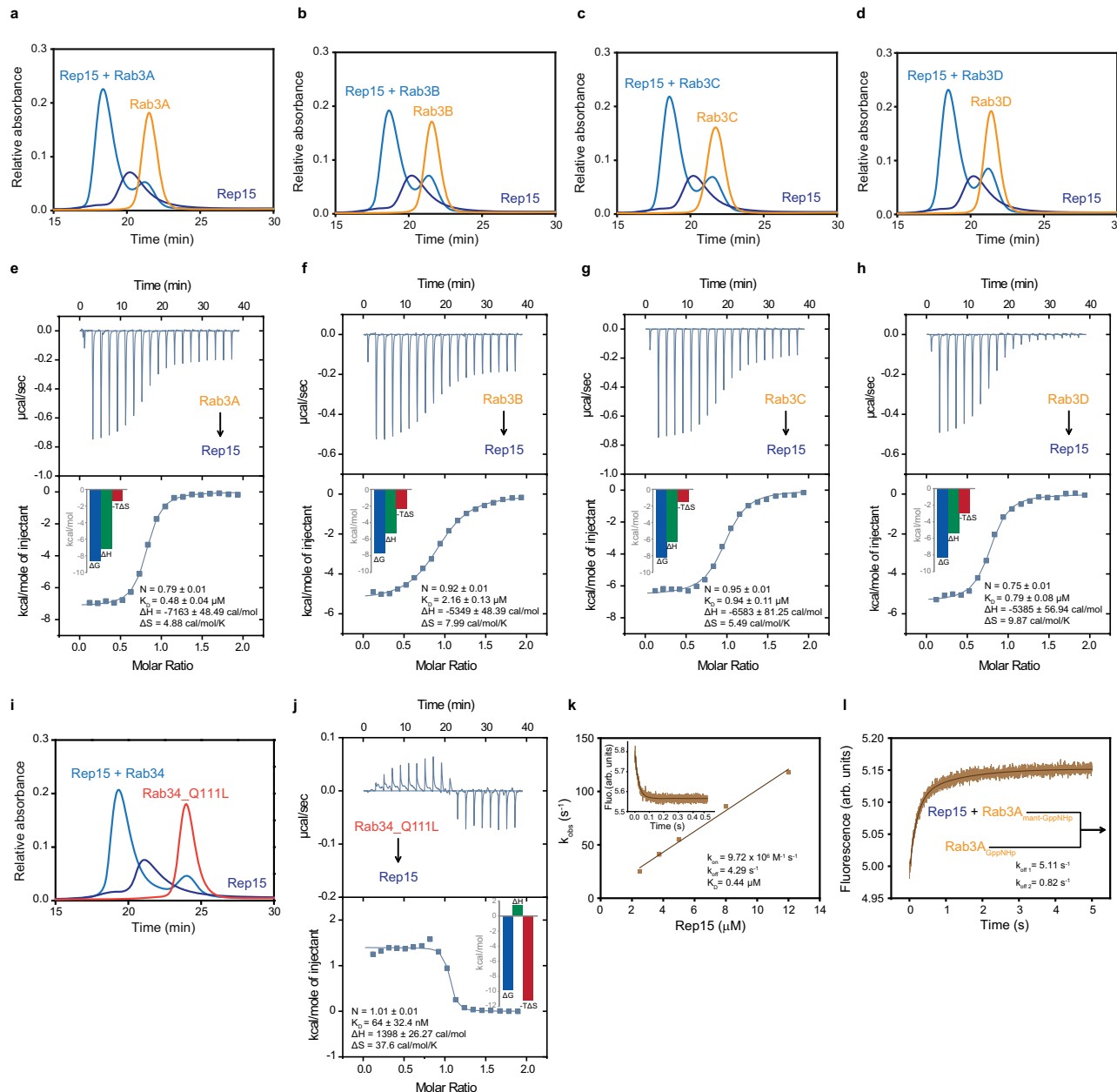

**Fig. 3 Quantitative analysis of the Rep15:Rab interaction. a–d** The Rep15 protein (110 μM, blue), the different Rab3 paralogs (121 μM, orange) and the mixture of both (sky blue) were subjected to an analytical size exclusion chromatography (aSEC) and tested for complex formation. A clear complex formation was detected. The curves shown are representative of at least three experiments. **e–h** Binding affinities were measured by titrating the Rab GTPase (500 μM) to full-length Rep15 (50 μM). Integrated heat peaks were fitted to a one-site-binding model yielding the binding stoichiometry (N), the enthalpy (ΔH), the entropy (ΔS), and the dissociation constant ($K_D$). The data are representative of at least three repetitions. **i** Interaction of Rep15 (110 μM, blue) with Rab34$_{Q111L\_1-237}$ (121 μM, red) was analyzed by analytical size exclusion chromatography. **j** ITC analysis of Rab34$_{Q111L\_1-237}$ (600 μM) binding to full-length Rep15 (60 μM). **k** Association rate constants ($k_{obs}$) for the Rep15 interaction were plotted against the increasing concentration of Rep15 (2–14 μM). Association was monitored by the change in fluorescence using a stopped-flow apparatus at 25 °C. Association of the Rep15 with Rab3A$_{mantGppNHp}$ leads to a decrease in fluorescence intensity and data is fitted to a single exponential. The change in fluorescence of 0.5 μM Rab3A$_{mantGppNHp}$ upon 3.75 μM Rep15 binding is shown in the inset. **l** Dissociation of Rab3A from the Rep15 was determined by monitoring the increase of fluorescence after mixing a complex of Rep15:Rab3A$_{mantGppNHp}$ (2 μM) with a 25-fold excess of non-fluorescent Rab3A$_{GppNHp}$. Data analyzed are averages of 6–8 individual traces.

was observed for Rab34 and the process is entropy-driven (Fig. 3j). Further, we measured association ($k_{on}$) and dissociation rate ($k_{off}$) constants for the Rep15 and Rab3A interaction using a stopped-flow apparatus. 0.5 μM mant-GppNHp Rab3A was used against increasing concentrations of Rep15 and the association of Rep15 leads to a decrease in fluorescence intensity of mant-GppNHp Rab3A. The observed $k_{obs}$ values were plotted against

increasing concentration of the Rep15 and an association rate constant of $9.72 \times 10^6$ M$^{-1}$ s$^{-1}$ was observed. A dissociation rate constant of 4.29 s$^{-1}$ was obtained from the intercept on the Y-axis (Fig. 3k). Next, we measured the dissociation rate constant directly, by mixing Rep15:mant-GppNHp Rab3A was rapidly mixed with a 25-fold excess of GppNHp Rab3A. Similar to Rab15, dissociation of Rab3A was biphasic, with a faster

dissociation rate constant of 5.11 s$^{-1}$ ($k_{off1}$) and a slower dissociation rate constant of 0.82 s$^{-1}$ ($k_{off2}$) was observed (Fig. 3l), and again the faster dissociation rate constant was similar to the rate constant measured indirectly. $K_D$ (0.44–0.53 μM) for the complex was calculated as the ratio of $k_{off}/k_{on}$, which is similar to the value obtained from ITC measurements.

Since we were not able to obtain crystals of the Rep15:Rab15 complex, we tried to crystallize the Rep15:Rab3 paralog complexes as well as the Rep15:Rab34 complex. To do this, we either mixed purified Rep15 and Rabs at 1:1 ratio or co-expressed and purified Rep15 ΔN1:Rab3A$_{Q81L\_19-217}$, Rep15 ΔN1:Rab3B$_{Q81L\_18-190}$ and Rep15 ΔN1:Rab34$_{Q111L\_1-237}$ complexes. Apart from human Rabs, we have also purified mouse Rab3A and bovine Rab3C and formed 1:1 complex with the human Rep15. All obtained complexes were used to perform the crystallization screen and finally, we obtained crystals for Rep15:Rab3C$_{GppNHp\_Q222H\_10-227}$ (bovine), Rep15:Rab3B$_{GppNHp}$ and Rep15_ΔN1:Rab3B$_{Q81L\_18-190}$ complexes.

**Overall architecture of the Rep15:Rab3C and Rep15:Rab3B complexes.** To understand the molecular basis of the interaction between Rep15 and Rabs, we have determined the structure of the Rep15:Rab3C$_{GppNHp\_Q222H\_10-227}$ (bovine) complex to a resolution of 2.52 Å, with the structure of Rab3A[35] (PDB: 3RAB) as a search model using molecular replacement and Phenix autobuild[36]. A partial model was obtained that was manually built and refined to convergence. The structure is composed of one Rep15:Rab3C$_{GppNHp\_Q222H\_10-227}$ complex per asymmetric unit. The crystallographic data collection and structure refinement statistics are provided in Supplementary Table 1.

The structure of Rep15:Rab3C reveals that, unlike most Rab effectors[8], Rep15 is a single domain globular protein which presumably explains why the other three deletion Rep15 constructs were not soluble. No electron density was observed for the first 14 amino acids, the middle 117 to 130 amino acids and for the last residue. Rep15 is composed of 7α-helices and 7β-strands. The central core is made by the β-strands which are flanked by the α-helices (Fig. 4a, b and Supplementary Fig. 3). No electron density was observed for the first 16 amino acids and the last 30 residues of Rab3C$_{GppNHP\_Q222H\_10-227}$ (bovine). A DALI search[37] against the PDB database revealed that the Rep15 fold shares some structural similarity with a Holliday junction resolving enzyme[38] (Hjc, PDB: 2EO0) with an RMSD of 2.9 Å for 108 aligned cα residues (Fig. 4c and Supplementary Figs. 3 and 4). The topologies of Rep15 and HJC are quite similar but some secondary structure elements have been changed from sheet to helix and vice versa. Still, both proteins share a similar (super) fold. However, the 7α + 7β arrangement is unique to Rep15 (Supplementary Fig. 3). Next, we compared our Rep15 structure with a model generated by the recently developed deep learning-based structure prediction method AlphaFold[39] and the prediction is quite accurate, showing an RMSD of 0.4 Å for all aligned Cα residues (Supplementary Fig. 3c). Rab3C is, expectedly, similar to Rab3A[35] (PDB:3RAB) with an RMSD of 0.7 Å for 168 aligned Cα residues.

Similar to all known Rab:effector complexes, switch I, switch II and the interswitch regions of Rab3C are at the interface with the effector. They interact with the α3 and α4 helices of Rep15, constituting the primary binding site, with a second site being composed of a loop (residues 115 to 145), ß2 and β3 of Rep15. α3 of Rep15 interacts with the switch I and interswitch regions, while α4 and β3 interact with the switch II region of Rab3C. Analysis of the interface by PDBePISA[40] revealed that the Rep15:Rab3C complex has a total buried interface of 843.4 Å$^2$. The binding interface has a central hydrophobic core that is supported by

several polar interactions. The hydrophobic patch is composed of switch I (V63 and I65), interswitch residues (F67 and W84) and switch II residues (Y92, I95 and Y99) that interact with the hydrophobic side chains of L88$^{Rep15}$, F89$^{Rep15}$ from α3 and F151$^{Rep15}$, L154$^{Rep15}$ and I155$^{Rep15}$ from α4. The side chain of L88$^{Rep15}$ forms an extensive hydrophobic interaction with a conserved triad of aromatic amino acids (F67, W84 and Y99), which are involved in all Rab:effector interactions[41]. The side chain of I94$^{Rep15}$ interacts with V63 of switch I. The V135$^{Rep15}$ hydrophobic side chain interacts with F67 while W140$^{Rep15}$ hydrophobic side chain interacts with W84 and Y99. Several polar interactions were also observed in Rab3C and Rep15, including those between side chains D66-T82$^{Rep15}$, D66-H84$^{Rep15}$, R91-Q107$^{Rep15}$, R91-D158$^{Rep15}$, Y92-Q107$^{Rep15}$, while the backbone of F67 forms a hydrogen bond with the side chain of H84$^{Rep15}$ and the backbone of V63 and I65 form hydrogen bonds with the side chain of Q85$^{Rep15}$. Sidechains of the interswitch residues K80 and Q82 form hydrogen bonds with the backbone of R134$^{Rep15}$ (Fig. 4a).

Subsequently, we also obtained Rep15:Rab3B$_{GppNHp}$ and Rep15 ΔN1:Rab3B$_{Q81L\_18-190}$ complex crystals, which diffracted to a resolution 2.75 Å and 2.8 Å, respectively and crystallized in space group C2 and P2$_1$ respectively. We solved the Rep15:Rab3B complex structure using Rep15:Rab3C (bovine) as a MR model (Supplementary Fig. 5 and Supplementary Table 1) and Rep15 ΔN1:Rab3B$_{Q81L\_18-190}$ complex structure using Rep15:Rab3B$_{GppNHp}$ complex structure as a MR model. The asymmetric unit contained two copies of the complex. Both copies share nearly the same overall architecture, as indicated by an RMSD of 0.124 Å for Rep15:Rab3B$_{GppNHp}$ complex and and an RMSD of 0.109 Å for the both copies of Rep15 ΔN1:Rab3B$_{Q81L\_18-190}$ complex (Supplementary Figs. 5, 6 and Supplementary Table 1). As expected, the structure of the Rep15:Rab3B complex is virtually identical to the Rep15:Rab3C complex structure with an RMSD of 0.272 Å for 302 aligned Cα residues (see below).

**Generation of Rep15 interface mutants.** The Rab-interacting surface of Rep15 is composed of residues 82–158. Most of the Rep15 interface residues are highly conserved in different mammalian homologs (Fig. 5a–c). Rep15 is quite conserved across the species except for the loop region connecting the β3 and α4 (Supplementary Fig. 7). To understand the importance of individual interface residues, we introduced structurally-guided mutations into Rep15 and monitored their interaction via ITC experiments. All conserved Rep15 mutants except for H84A show a significant decrease in binding affinity. Double alanine mutations of H84 and Q85 abrogate Rab3A interaction (Fig. 5d, e). H84 forms polar interactions with the crucial D66$^{Rab3C}$ and a hydrogen bond with the backbone of F67$^{Rab3C}$ while Q85 form hydrogen bonds with the backbone of several amino acids. Next, to identify the importance of each amino acid, we mutated H84 and Q85 to alanine and could show that Q85 is essential for Rab3A interaction and Q85A mutation led to more than a 30-fold reduction in affinity, whereas mutation H84A does not have any effect on the interaction (Fig. 5f, g). Mutation of L88A leads to a 20-fold reduction in the binding affinity, with the L88 side-chain inserting itself into the hydrophobic pocket created by the conserved aromatic triad (F67, W84 and Y99) (Fig. 5h). The mutations Q107A and L154A_I155A led to a complete loss of interaction (Fig. 5i, l). Q107 forms hydrogen bonds with R91$^{Rab3C}$ and Y92$^{Rab3C}$. L154 forms hydrophobic interactions with I95 whereas I155 interacts with Y92 and I95. The D158A mutation leads to a 12-fold reduction in binding affinity (Fig. 5m). To investigate the importance of non-conserved Rep15

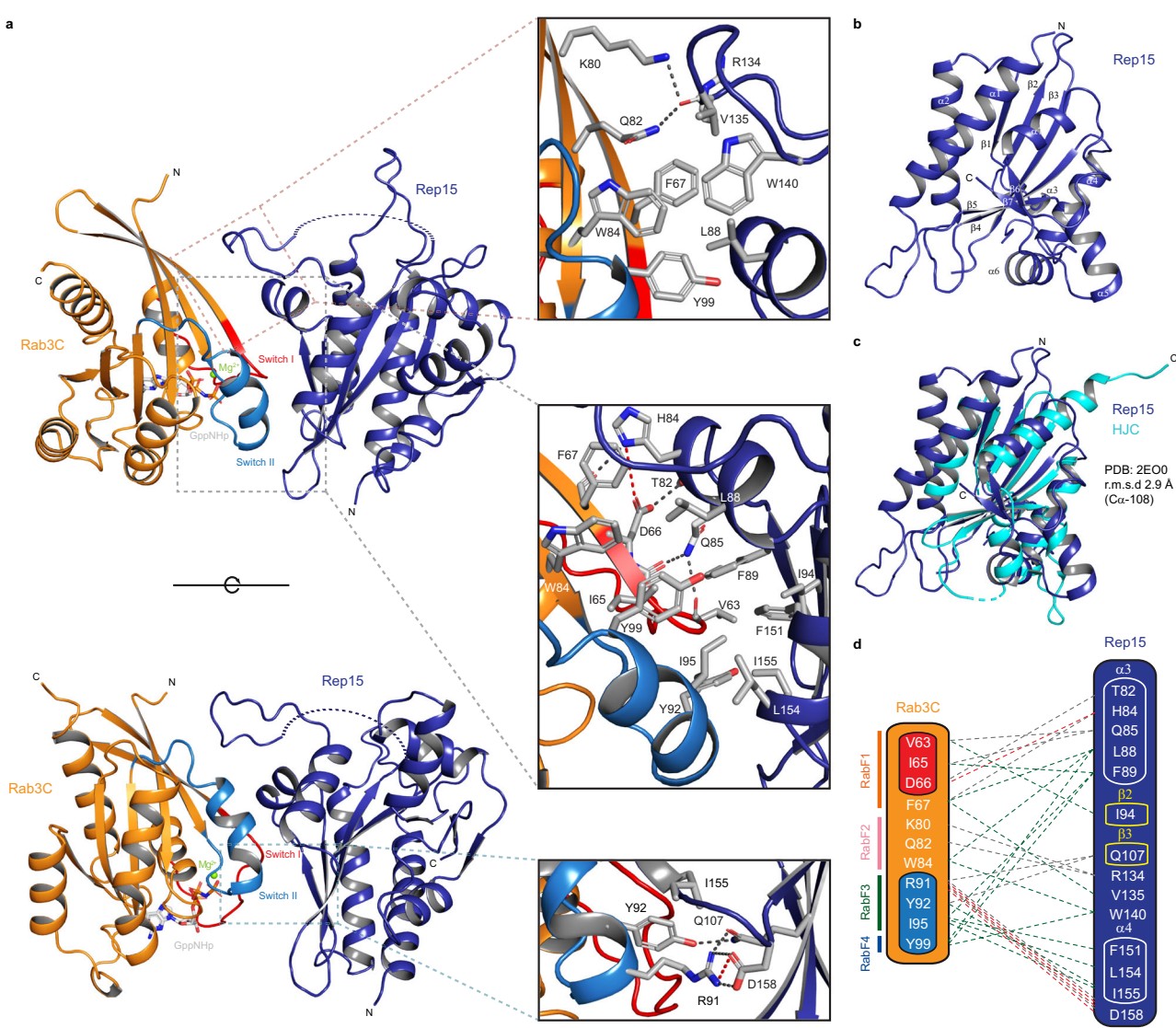

**Fig. 4 Crystal structure of the Rep15:Rab3C$_{GppNHp\_Q222H\_10-227}$ complex. a** Cartoon representation of the Rep15:Rab3C$_{GppNHp\_Q222H\_10-227}$ complex. Rab3C (orange) binds to Rep15 (blue) via its switch I, switch II and interswitch regions. Switch I and switch II regions are highlighted in red and sky blue, respectively. Mg$^{2+}$ ion is shown as a green sphere and GppNHp is shown in stick representation. Insets show the zoom-in on the binding interface of Rep15 and Rab3C$_{GppNHp\_Q222H\_10-227}$. Key interactions at the Rep15:Rab3C$_{GppNHp\_Q222H\_10-227}$ interface are shown in stick representation and labeled. **b** The Rep15 is composed of 7α-helices and 7β-sheets forming a novel α + β fold. **c** Structural superposition of Rep15 (blue) and *Sulfurisphaera tokodaii* HJC (cyan, PDB: 2EO0). **d** Schematic illustration of the Rep15:Rab3C binding interface. Hydrogen bonds and ionic interactions are shown in gray and red dashed lines, respectively. Green dashed lines indicate the hydrophobic interactions. RabF1, RabF2, RabF3, and RabF4 motifs are shown in gray, red, green, and purple, respectively.

loop residues in Rab interaction, we mutated V135 and W140 to alanine. However, this does not affect interaction with Rab3A (Fig. 5j, k).

Additionally, we also performed qualitative aSEC experiments to monitor the stability of mutant Rep15:Rab3A/Rab34/Rab15 complexes, and in agreement with the ITC results, Rep15 constructs having low Rab3A binding affinity failed to form stable complexes (Supplementary Figs. 8–10).

**Molecular basis of Rab selectivity.** To understand the molecular basis of Rep15 selectivity towards Rab15, Rab3 paralogs and Rab34, we aligned other evolutionarily similar Rab GTPases and observed that most of the Rep15 interacting residues are conserved (Fig. 6a, b and Supplementary Fig. 11). However, our Y2H experiments suggest that Rep15 does not interact with these evolutionarily similar Rabs, although the absence of a signal for

Rab15, which clearly does interact with Rep15, means that this result is not conclusive. To further validate the result, we checked the Rep15 interaction with Rab27A, which shares several effectors with Rab3 paralogs, but did not observe any interaction in aSEC experiments (Fig. 6d), supported by low binding affinity (Fig. 6i). Next, we also checked the interaction of Rep15 with Rab8A, which is evolutionary similar to Rab15 and both Rab8A and Rab15 are known to interact with bMERB family effectors[15], but no interaction was observed (Fig. 6c). Sequence alignment shows that V69$^{Rab3C\_bovine}$ (V61$^{Rab3A\_human}$) of switch I is replaced by a charged/polar residue in evolutionarily similar Rabs (Fig. 6b). However, this residue is not directly involved in any interaction and lies next to the interacting residues. Y92$^{Rab3C\_bovine}$ (Y84$^{Rab3A\_human}$) of switch II residue is another non-conserved residue that interacts with the crucial Q107$^{Rep15}$ and also interacts with I155$^{Rep15}$. This tyrosine residue is replaced by phenylalanine in other Rab8 family members (Rab8A/B, Rab10 and

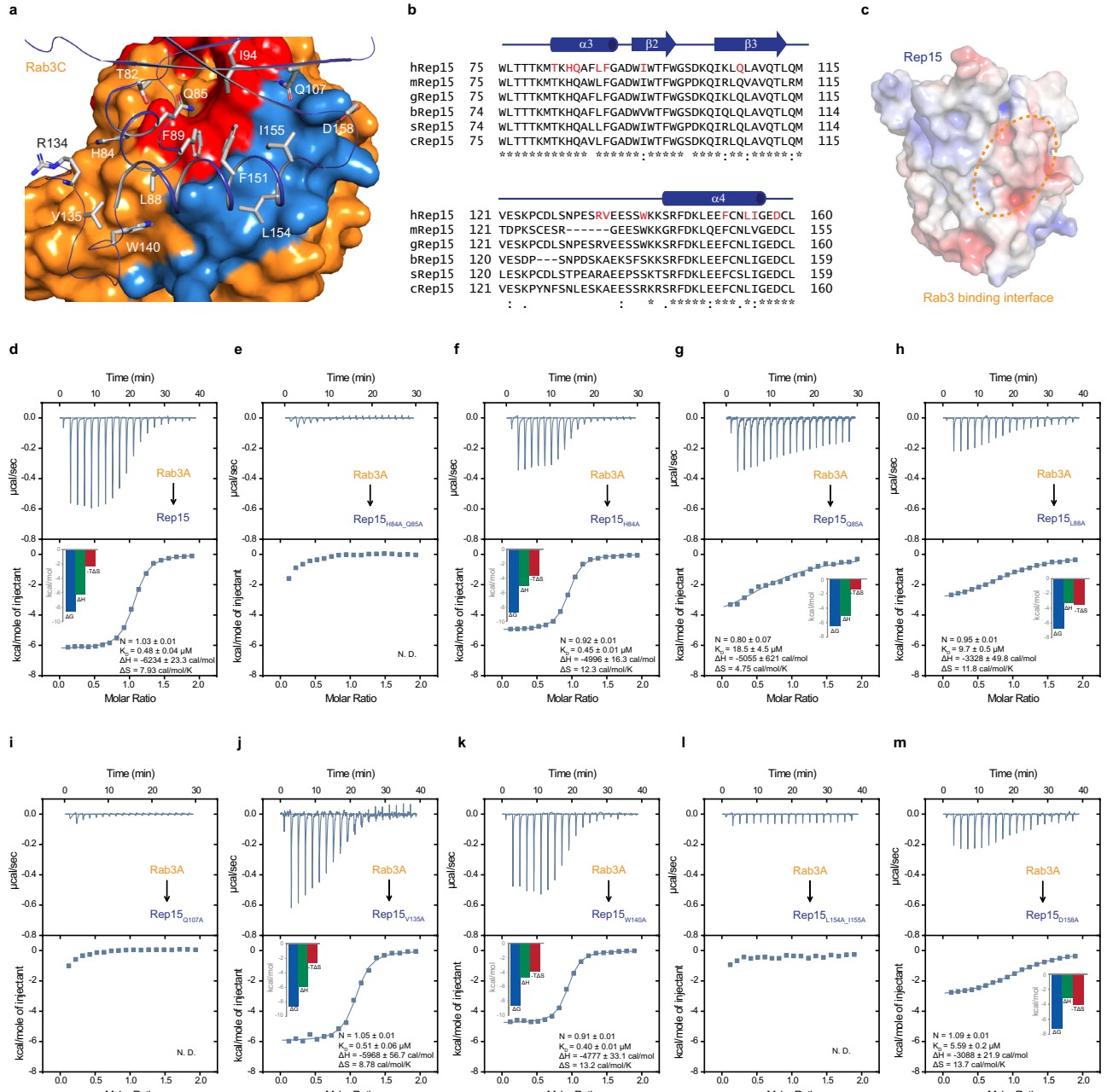

**Fig. 5 Crucial Rep15 interface residues for Rab interaction. a** Zoomed in the overview of Rep15:Rab3C$_{GppNHp\_Q222H\_10-227}$ binding interface. Key Rep15 residues involved in the interaction with Rep15 are shown as sticks and labeled. Rab3C$_{GppNHp}$ interaction interface is shown as surface representation. Switch I and switch II regions are shown in red and sky blue, respectively. **b** Sequence alignment of the Rab binding site of human Rep15 along with mammalian homologs from *Mus musculus* (mRep15), *Gorilla gorilla gorilla* (gRep15), *Bos taurus* (bRep15), *Sus scrofa* (sRep15) and *Canis lupus familiaris* (cRep15). Residues involved in Rab3 interaction are labeled in red. **c** Electrostatic potential of the Rep15 calculated in PyMOL using the APBS-PDB2PQR plugin and visualized in red to blue (−5 kT/e to +5 kT/e). The dashed line highlights the Rab3 binding interface. **d–m** Alanine screening of the Rab binding interface residues of Rep15. The binding of Rab3A$_{GppNHp}$ with different Rep15 mutants was systematically tested and affinities were determined by ITC experiments. Mutation of H84_Q85 and Q107 to alanine led to the loss of interaction. L88 which interacts with the conserved aromatic triad of Rab3C (F67, W84 and Y99), led to a defect in Rep15 interaction. Whereas in the case of L154_I155A mutant of the Rep15 hydrophobic patch, no interaction was detected. N.D. denotes not detected. The data are representative of at least three repetitions.

Rab13) and also in other exocytic Rabs (Rab26, Rab27A/B and Rab37). However, surprisingly in Rab34, this tyrosine residue is replaced by phenylalanine and a lysine residue replaces the neighboring arginine. Besides the changes in the switch II region, Rab34 has an extended N-terminus that may assist Rep15 interaction. To assess the significance of this tyrosine residue, we purified Rab27A_F81Y/Rab3A_Y84F/Rab34_Q111L_F114Y and checked the interaction with Rep15. Compared to wild-type

Rab27A, the mutant F81Y Rab27A shows some complex formation with Rep15 in aSEC experiment and compared to wild type, the F81Y mutant shows more than an 8-fold increase in Rep15 binding affinity (Fig. 6d, e, i and j). On other hand, Rab3A Y84F mutant shows a 23-fold reduction in Rep15 binding affinity, whereas the V61N mutation does not affect Rep15 interaction (Fig. 6i–m). The Y84F$^{Rab3A}$ mutant failed to form a stable complex with Rep15 in an aSEC experiment (Fig. 6g–h) Thus, our

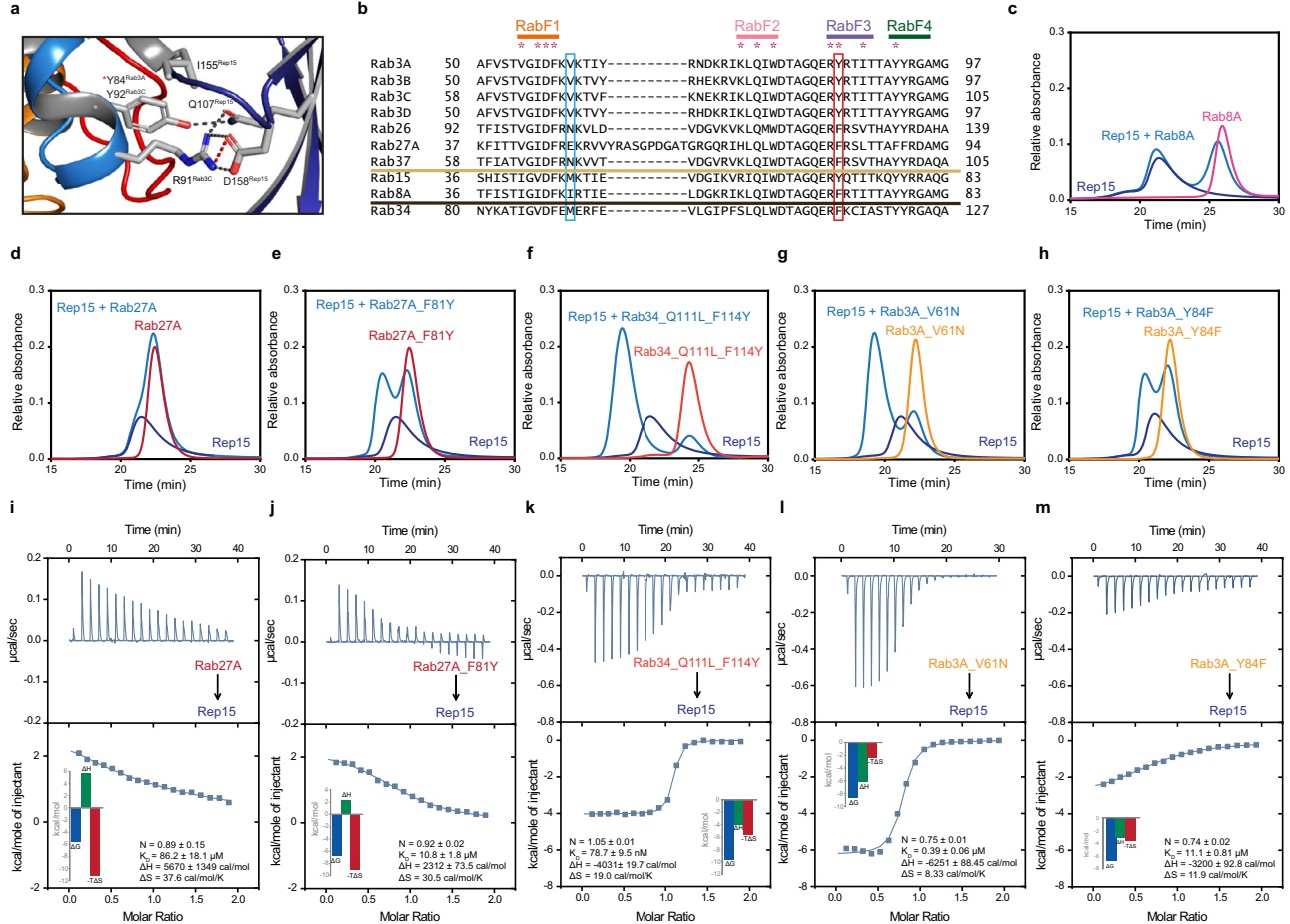

**Fig. 6 Tyrosine within the RabF3 motif of Rab3 determines specificity towards Rep15. a** The zoom-in on the binding interface of Rep15:Rab3C$_{GppNHp\_Q222H\_10-227}$ (bovine) complex structure. * indicates the position of equivalent tyrosine in Rab3A. **b** The sequence alignments of different evolutionary similar human Rab proteins clearly show that most of the Rep15 interacting residues are conserved except for the residues indicated in the sky blue and red bar. Residue V61 lies next to the binding interface whereas Y84 is interacting with crucial Q107$^{Rep15}$ and D158$^{Rep15}$. **c–h** The Rep15 protein (110 μM, blue), the different Rabs (121 μM, orange) and the mixture of both (sky blue) were subjected to an analytical size exclusion chromatography (aSEC) and tested for complex formation. Rab8A and Rab27A do not form a stable complex. However, Rab27A$_{F81Y}$ mutant shows some complex formation with the Rep15. V61N mutation of Rab3A and F114Y mutation or Rab34 does not affect its interaction. with Rep15. **i–m** Binding affinities were measured by titrating the Rab GTPase (500 μM) to full-length Rep15 (50 μM). Integrated heat peaks were fitted to a one-site-binding model yielding the binding stoichiometry (N), the enthalpy (ΔH), the entropy (ΔS), and the dissociation constant (K$_D$). For Rab27A, Rab27A$_{F81Y}$ and Rab34$_{Q111L\_F114Y\_237}$ titration to full-length Rep15, 600 μM Rab is titrated to 60 μM Rep15. N.D. denotes not detected. The data are representative of at least three repetitions.

result explains why Rab8 family members and other exocytic Rabs are not able to bind Rep15. Rab34_Q111L_F114Y mutant forms a stable complex with Rep15 with a similar affinity as of wild type (Fig. 6f, k). Further biochemical experiments are needed to explain the molecular basis of Rab34 interaction with Rep15.

**Comparison with previously solved Rab15, Rab3 and Rab34-effectors.** Most Rab effectors bind to their cognate Rabs by 1 or 2 α-helices[8], with a few exceptions such as ankyrin repeat containing effectors like VARP (interacts with Rab32 via 4 parallel helices)[42], ACAP2 (interacts with Rab35 via 3 parallel helices)[43], the RUN domain of RUSC2 (interacts with Rab35 via 3 helices)[43], MyoVb (binds to Rab11a via 3 helices and a loop)[44], while the Rab5 binding interface of EEA1 is composed of a short β-hairpin and an α-helix[45], whereas the OCRL Rab8 binding interface is made up of an Immunoglobulin-like β-strand domain and an α-helix[46].

In contrast with previous studies suggesting that most of the Rab15[15,16], Rab3 paralogs[47] and Rab34[48] effector domains are made up of 1-3 α-helices, Rep15 is a globular effector with a

7α + 7β arrangement (Fig. 7a and Supplementary Fig. 12). Unlike other Rab3 effectors such as Rabphilin-3A[21], Noc2, Slp4 which are known to interact with Rab27, Rep15 is quite specific to Rab3 paralogs and does not interact with other secretory Rabs. Compared to the Rep15:Rab3C$^{bovine}$ binding interface (843.4 Å$^2$), the Rabphilin-3A:Rab3A complex[47] (PDB: 1ZBD) structure has a larger binding interface (1426.5 Å$^2$) (Fig. 7a–b). The Rabphilin-3A:Rab3A complex has two distinct binding interfaces, the first site being made up of switch I, interswitch and switch II region and the second site is made up of RabCDRs (Complementary determining region) (Fig. 7d). Structurally-based sequence alignment of Rab3A and Rab3C shows that apart from the switch I, interswitch and switch II residues, three RabCDRs of Rab3A are involved in making contacts with the Rabphilin-3A, whereas in the case of the Rep15:Rab3 complex structures, RabCDRs are not part of the binding interface (Fig. 7e–f and Supplementary Fig. 12a). Raphilin-3A has a structural element SGAWFF motif which interacts with the RabCDRs (Supplementary Fig. 12b). Similar structural elements (S/T)(G/L)xW(F/Y)$_2$ are present in other Rab27A/B effectors, for example, Slac-2a has a SLEWYY motif and Slp-2a has a SGQWFY motif. However,

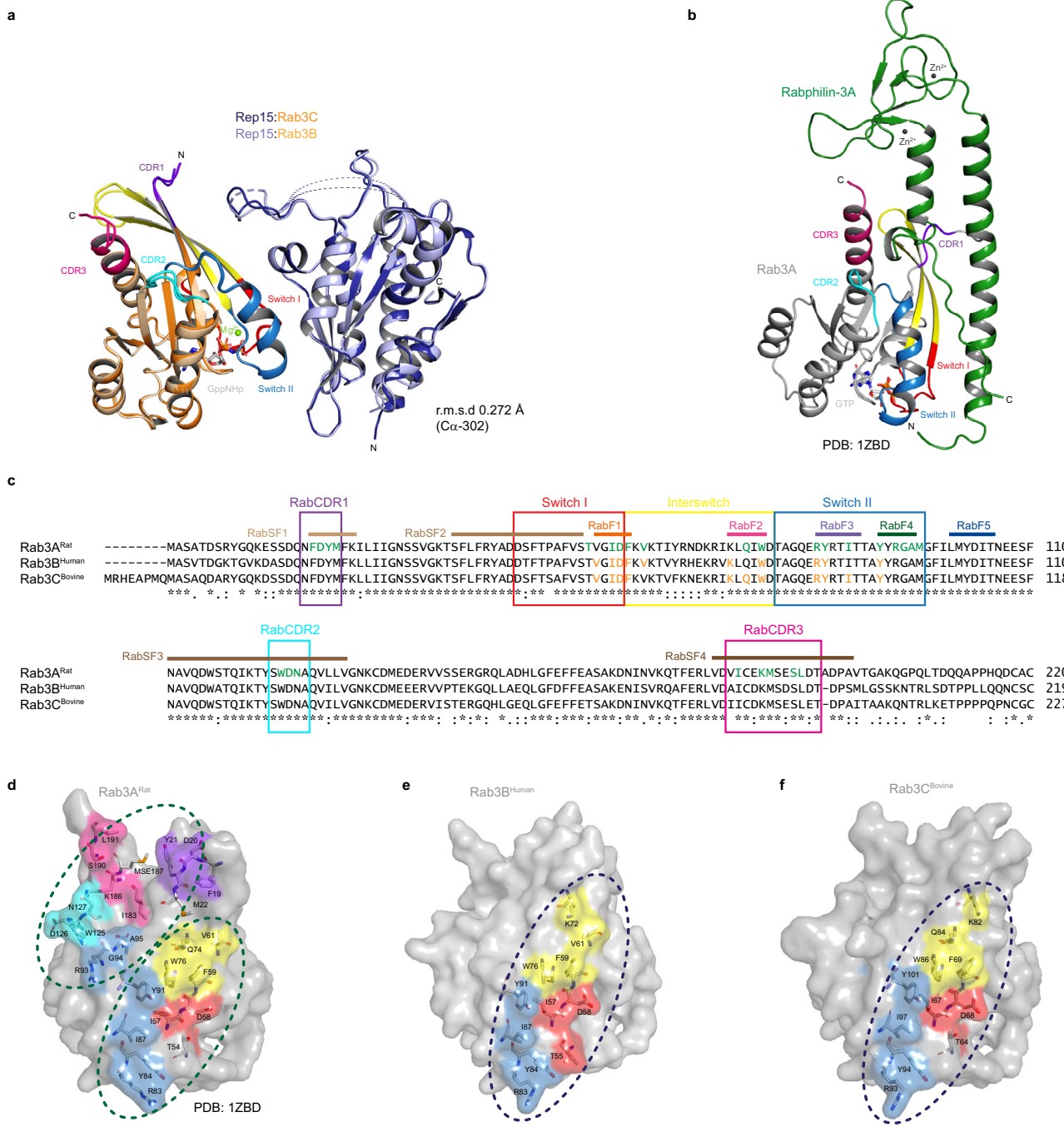

**Fig. 7 Comparison of the Rep15:Rab3B_GppNHp/Rab3C_GppNHp complex with the Rabphilin-3a:Rab3A complex structure. a** Superposition of the Rep15:Rab3B_GppNHp complex and Rep15:Rab3C_GppNHp_Q222H_10-227 complex structure. **b** Cartoon representation of Rabphilin-3a:Rab3A complex (PDB:1ZBD). Switch I, interswitch and switch II regions are highlighted in red, yellow and sky blue, respectively. Mg$^{2+}$ ion is shown as a green sphere and GppNHp/GTP is shown in stick representation. CDR1, CDR2 and CDR3 regions are highlighted in purple, cyan and pink, respectively. **c** sequence alignment of rat Rab3A, human Rab3B and bovine Rab3C. Switch I, interswitch and switch II regions are shown in red, yellow and sky blue boxes, respectively. CDR1, CDR2 and CDR3 regions are shown in purple, cyan and pink box, respectively. Identical residues are denoted by * and similar residues by:. Rabphilin-3a and Rep15 interacting residues are marked in green and orange color, respectively. **d–f** A detailed comparison of the effector interaction surfaces of rat Rab3A, human Rab3B and bovine Rab3C. Rabs are shown as gray surfaces and effector interacting residues are shown as sticks.

Rep15 lacks such a motif. Rab effector specificity is defined by the Rab family motifs (RabF 1-5) and Rab subfamily motifs (RabSF 1-4)[49]. RabF motifs cluster in and around switch regions whereas the RabSF1 motif overlaps with the RabCDR1, RabSF3 to RabCDR2 and RabSF4 to RabCDR3 region[49] (Fig. 7c and Supplementary Fig. 11). The Rab27 specific effector Slac2-a/

melanophilin shares a similar domain architecture with Rabphilin-3A (Supplementary Fig. 12b, c) and Slac2-a specificity is provided by the Rab27 CDR1 region/RabSF1 and small changes in switch II regions[50]. Slp-2a has some similarities with Raphilin-3A and Slac2-a but lacks Zn$^{2+}$ fingers (Supplementary Fig. 12d). In the case of Rep15 binding, the Rab3 paralogs/Rab15 specificity

is determined by the RabF3 motif, which has a conserved crucial tyrosine residue and phenylalanine in evolutionarily similar Rabs (Supplementary Fig. 11). We have shown that the Rab3A Y84F mutation leads to a 20-fold reduction in Rep15 binding affinity and conversely, Rab27A F81Y forms a complex with Rep15. However, this complex is weaker compared to Rep15:Rab3A, but this can be explained by the fact that the Rab27A RabF4 (**F**FRDG) motif has phenylalanine whereas Rab3A has a tyrosine residue (Y91; RabF4: **Y**YRGA) which is also part of the binding interface (Supplementary Fig. 11). Taken together, we propose that the tyrosine residue of the RabSF3 motif (R**Y**RTIT) provides specificity in the case of Rab3 paralogs and Rab15.

Notably, Rab34 has phenylalanine instead of tyrosine of RabF3 motif (R**F**KCIA) and still interacts with Rep15. This can be explained by the fact that Rab34 is evolutionarily distant from the other Rep15 interacting Rabs and the Rab34 switch II region is quite different (Fig. 6b and Supplementary Fig. 11) and also has a unique long N-terminal extension, suggesting that apart from switch I/II, another Rab34 binding interface could be involved for Rep15 interaction, which is supported by the fact that the F114Y mutation of Rab34 does not lead to an increase in Rep15 binding affinity. This would mean that the same effector molecule can bind to different Rabs via a different binding interface. To confirm the binding mode for Rab34, further biochemical mutagenesis and structural experiments are needed. Apart from Rep15, Rab34 is also known to interact with another effector, RILP, and during ciliogenesis in hTERT-RPE1 cells, the unique long N-terminal region of Rab34 was shown to be essential and not, as usual, the specific switch II region[51] and N-terminal LPQ motif which are otherwise necessary for ciliogenesis[52]. However, biochemical and structural data describing the RILP:Rab34 interaction is missing. RILP has been shown to interact with Rab7A and the RILP:Rab7A complex is required for endosomal transport to lysosomes and the structure of RILP:Rab7A[48] (PDB: 1YHN) is available (Supplementary Fig. 12e). For the RILP:Rab7A interaction RabSF1 (RabCDR1) and RabSF4 motif within the switch II regions provide specificity[48]. For Rab15 effector molecules (Supplementary Fig. 12f–h), we have previously reported that the N-terminal of Rab8 (RabSF1 motif/RabCDR1) is crucial for the interaction with the bMERB family effectors[15].

**Rep15 depletion results in decreased cell proliferation, delayed wound healing and inhibits anchorage-independent growth.** To explore the physiological function of Rep15, we looked for a cell line with maximal Rep15 expression in the Human Protein Atlas database[31]. At the time of the study, we found U138MG glioblastoma cells to have the highest expression of Rep15 among tested cell lines. CRISPR-Cas9 was used for the generation of the Rep15 KOs, and the individual clones were validated via PCR by primer pairs flanking the two guide RNAs. A complete absence of the respective amplicon indicated the generation of KO clones (Supplementary Fig. 13a–b). Further, the expression of Rep15 mRNA was measured by quantitative real-time PCR, resulting in no signal corresponding to Rep15 mRNA in the KO cells with a subsequent increase in the rescue cell lines (Supplementary Fig. 13c). Cell proliferation was measured by seeding an equal number of cells and counting them for six days. The KO cells (KO) significantly showed decreased cell proliferation compared with the corresponding Cas9 control cells (UC) (Fig. 8a). The two rescue cell lines (KO^WT and Rab-binding deficient mutant KO^Q107A) both showed a significant increase in proliferation compared to the KO cells (Fig. 8a). This result might indicate a pro-tumorigenic role of Rep15, at least in the tested cell line. The fact that both the WT and Q107A mutant showed similar rescue

in the phenotype would indicate a Q107-independent role of Rep15 in the regulation of cell proliferation. This would need a detailed characterization of the role of Rep15 in cell proliferation, which is beyond the scope of this manuscript.

Aberrant expression of Rab proteins has been reported in multiple cancers such as lung, brain and breast malignancies[30]. Mutations in Rab-coding genes and/or post-translational modifications in their protein products disrupt the cellular vesicle trafficking network modulating tumorigenic potential, cellular migration and metastatic behavior[53]. To this end, we measured whether deletion of Rep15 affected the expression of any of the interacting Rabs. Interestingly, some of the interacting Rabs (Rab3B, Rab15 and Rab34) were significantly downregulated, while the expression of Rab3A remained unchanged (Fig. 8b). We could not detect the expression of Rab3C and Rab3D in U138MG cells. It is worth noting that both Rab15, Rab3 paralogs and Rab34 have been suggested to have pro-tumorigenic potential[13,54–56]. We also measured the expression of Rab8A since it is evolutionarily similar to Rab15 and Rab27A as it shares many effector molecules with Rab3 and has been shown to be highly expressed in synaptic vesicles. We did not find any expression differences in these Rabs, indicating the effect might be restricted to interacting Rabs only. In the two rescue cell lines (KO^WT and KO^Q107A), there was a slight trend in the upregulation of the expression of the Rab3B and Rab34, which was only significant in the case of Rab3B for the rescue with the WT construct (Fig. 8b). Further, we did not find any change in the expression of other Rabs (Rab15, Rab8A and Rab27A) in the rescue cell lines compared to the Rep15 KO cells (Fig. 8b). We also measured the migratory potential of the control, Rep15 KO and the rescue cell lines via a wound-closure assay. Both KO and rescue cell lines (KO^WT and KO^Q107) showed diminished migration over the 48 h period that was sufficient for the Cas9 control cells (UC) to completely close the wound (Fig. 8c, d).

One of the hallmarks of tumor cells is that they can survive and grow in the absence of anchorage to the extracellular matrix (ECM) and their neighboring cells. This property, termed as anchorage independence of growth, correlates closely with tumorigenicity[57]. Deletion of Rep15 significantly decreased the number of colonies that could grow in an anchorage-independent fashion as compared to the Cas9 control (UC) (Fig. 8e, f). This phenotype could also be partially rescued by the overexpression of either the WT or Rab binding deficient mutant (Q107A) (Fig. 8e, f).

**Rep15 deletion impairs receptor recycling.** Since Rep15 has been implicated in facilitating receptor recycling from the endocytic recycling compartment (ERC)[18], we performed transferrin (Tfn) uptake and recycling assays in the control, Rep15 KO and the two rescue cell lines (Flag-tagged stable cell lines, mixed population). We could show that recycling of transferrin was significantly reduced in KO cells compared to control, indicated by significant retention of transferrin inside the cells (Fig. 9a–c; UC vs. KO). This could be rescued by overexpression of the WT Rep15 in the KO cells (Fig. 9a–c; KO vs. KO^WT). Overexpression of Rep15 in the WT cells also resulted in a significant decrease in the recycling of transferrin similar to the KO cells (Fig. 9a–c; UC and UC^WT). A similar observation was made by Strick et al.[18]; this could be due to the titration of other partners involved in endocytic steps. Next, we repeated the above measurements in the KO and the two rescue cell lines to see the effect of the Rab binding deficient mutant (Q107A) in this process. The WT Rep15 (KO^WT) could rescue transferrin recycling in the KO cells while overexpression of Q107A mutant (KO^Q107A) did not affect this process (Fig. 9d–f). To further validate our finding, we transiently

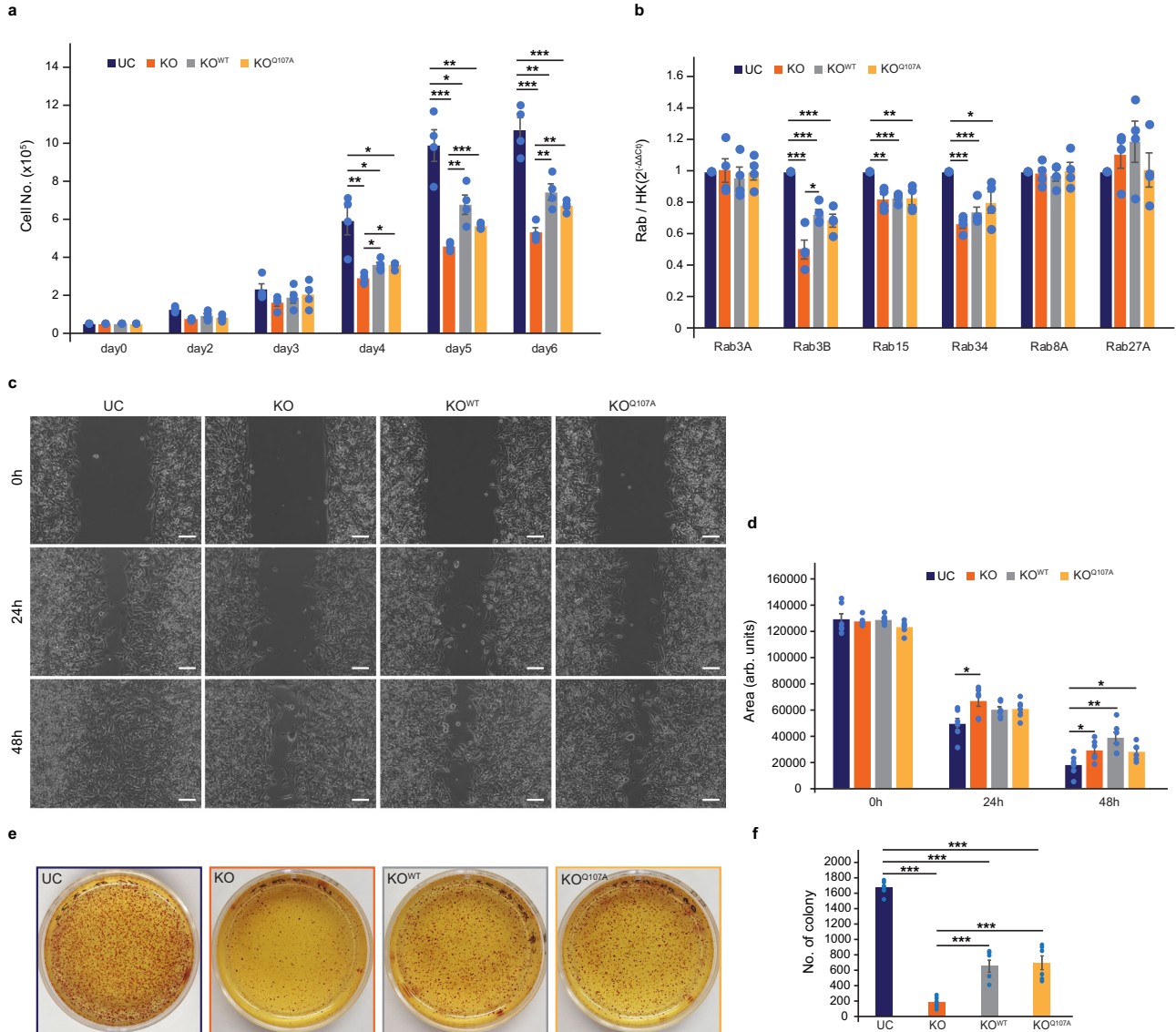

**Fig. 8 Rep15 KO U138MG cells show reduced cell proliferation, migration and anchorage-independent growth. a** Rep15 KO cells (orange bars) show a significant reduction in cell proliferation as measured by counting the number of cells for six days. Both the rescue cell lines (KO$^{WT}$ (gray bars) and Rab-binding deficient mutant KO$^{Q107A}$ (yellow bars) showed a significant increase in the cell proliferation over the knockout background ($n = 4$). **b** Real-time quantitative PCR analysis of the KO for the Rep15 indicates a significant reduction in the mRNA expression of Rab3B, Rab15 and Rab34, while the expression of Rab3A, Rab8A and Rab27A remain unchanged. The expression of measured Rabs in the two rescue cell lines did not look different from the KO cells, except in the case of rescue with the WT-Rep15 (KO$^{WT}$), for Rab3B ($n = 4$). **c** Rep15 KO cells show a reduced cell migration in a 2D scratch assay, with no significant difference in the two rescue cell lines (representative picture; 10× magnification). Scale bar, 100 μm. **d** Quantification of the 2D scratch assay ($n = 6$). **e** The KO cells show a reduced capacity to form colonies in the anchorage-independent growth, with the significant rescue with both the WT and Q107A mutant (representative picture). **f** Quantification of the number of colonies counted in **e** ($n = 6$). Data are presented as mean value ± SEM. *$p < 0.05$, **$p < 0.01$, ***$p < 0.001$, (unpaired, two tailed, Student's *t*-test). Source data are provided as a Source Data file. n represent the biological repeats. The exact *p* values are provided in the Source data file.

overexpressed the GFP-tagged WT or Q107A mutant in KO cells and measured transferrin recycling. The cells were grouped under low and highly-GFP expressing cells, with only the low-GFP cells showing rescue, while the highly-GFP expressing cells failed to rescue the recycling defect, further validating the fact that strong overexpression of WT Rep15 had an adverse effect on the recycling pathway even in the KO background (Supplementary Fig. 14a, b). The transient overexpression of the mutant did not look different from the KO cell line (Supplementary Fig. 14b). Further, we transiently overexpressed the GFP-tagged WT and Q107A mutant in the WT cells, the overexpression of WT Rep15 (UC$^{WT}$) significantly reduced the recycling pathway while Q107A

mutant (UC$^{Q107A}$) did not (Supplementary Fig. 14c, d), in line with the findings from Strick et al.[18] and also in agreement with our biochemical analysis of the mutant (Q107A), which shows minimal/no interaction with the Rab proteins. Since both the knockout and overexpression of Rep15 have a similar inhibitory effect on the recycling pathway the rescue experiments should be interpreted cautiously. Since we had mixed rescue phenotypes in different experiments, the physiological aspect of the effect of the Rab binding deficient (Q107A) mutant should be carefully validated in detail in future studies.

Taken together, our data provide a detailed overview of the Rep15:Rab interaction. Here, we have identified Rab3 paralogs

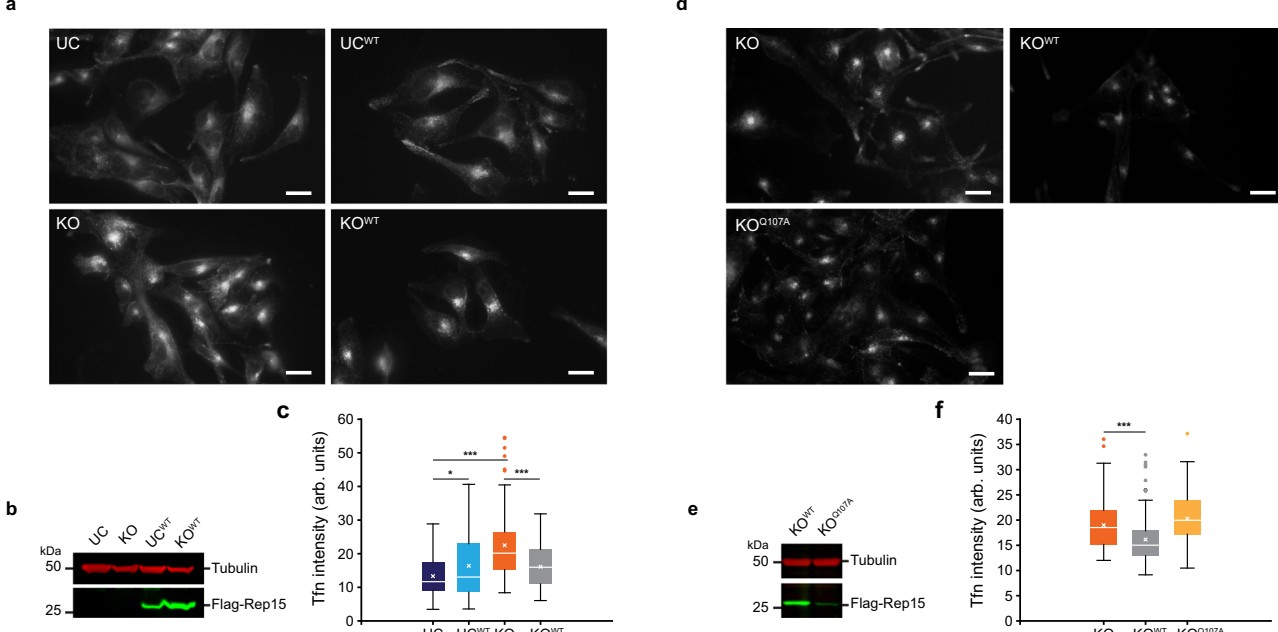

**Fig. 9 Rep15 KO U138MG cells show reduced transferrin recycling. a** Rep15 KO shows significant retention of Alexa-488-conjugated human transferrin in the 2 h chase post 30 min. internalization (representative pictures). Scale bar, 10 µm. Overexpression of the WT-Rep15 further increased the transferrin retention in the WT cells (UC$^{WT}$; light blue bar) while, decreasing it, in the Rep15 KO cells (KO$^{WT}$; gray bar). **b** Western blot confirmation of the overexpression of Flag-tagged WT-Rep15. **c** Quantification of retained transferrin in the respective cell line presented in **a** ($n = 60$; cells were examined over three independent experiments). **d** The rescued KO cells with the WT Rep15 (KO$^{WT}$; gray bar) show a significant reduction in the transferrin retention in the chase experiment, while the Q107A mutant (KO$^{Q107A}$; yellow bar) did not look different from the KO cells (representative picture). Scale bar, 10 µm. **e** Western blot confirmation of the two Flag-tagged constructs. **f** Quantification of the transferrin from the cells presented in **d** ($n = 100$; cells were examined over three independent experiments). Data are presented as box plot, showing mean (cross), median (line), 25$^{th}$ and 75$^{th}$ percentile (box). The whiskers extend to the most extreme data points not considered outliers, and the outliers are represented as dots. *$p < 0.05$, **$p < 0.01$, ***$p < 0.001$, (unpaired, two tailed, Student's t-test). Source data are provided as a Source Data file. n represent the number of cells analyzed for the transferrin assay from three biological repeats. The exact p values are provided in the Source data file.

and Rab34 as novel Rep15 interaction partners and thoroughly characterized the Rep15:Rab interaction. We report the first Rep15 structure in a complex with Rab3B/C. Further, our mutagenesis experiments shed light on the origins of Rep15 selectivity towards specific Rabs. Additionally, we have demonstrated that loss of Rep15 results in decreased cell proliferation and migration of glioblastoma cell lines. Apart from having a role in receptor recycling, comparative proteomics studies of *Mycobacterium bovis* infected human macrophages showed that Rep15 is one of the genes overexpressed upon infection[58]. Recently, Rab3 paralogs have been implicated in various cancers[56] and Rab34 is essential for ciliogenesis[59,60]. Taken together, this suggests that it would be important to study the role of Rep15 in cancer, ciliogenesis and *Mycobacterium bovis*-induced infection.

## Methods

**Plasmid cloning**. All the expression constructs used in this study were generated by standard cloning techniques, using Phusion polymerase, restriction digestion, and ligation by T4 DNA ligase. Point mutants were generated by quick-change site-directed mutagenesis, using Phusion polymerase. A detailed overview of all expression constructs employed in this study is presented in Supplementary Tables 2 and 3. All plasmids were verified by DNA sequencing.

**Antibodies**. ANTI-FLAG® antibody produced in rabbit F7425 (polyclonal, Sigma, https://www.sigmaaldrich.com/DE/en/product/sigma/f7425, dilution 1:1000) and anti-alpha-Tubulin monoclonal (DM1A) mouse IgG1 T9026 (Sigma, https://www.sigmaaldrich.com/DE/de/product/sigma/t9026, dilution 1:3000) were used as primary antibodies. IRDye® 800CW Goat anti-Rabbit IgG Secondary Antibody (LI-COR Biosciences, https://www.licor.com/bio/reagents/irdye-800cw-goat-anti-rabbit-igg-secondary-antibody, dilution 1:10000) and IRDye® 680RD Goat anti-Mouse IgG Secondary Antibody (LI-COR Biosciences, https://www.licor.com/bio/

reagents/irdye-680rd-goat-anti-mouse-igg-secondary-antibody, dilution 1:10000) were used as secondary antibodies.

**Yeast two-hybrid assay**. Yeast two-hybrid assay was performed, as described previously[61]. Briefly, pLEXA–Rab$_{QL}$ (Dominant active) constructs were transformed into the yeast strain Y187 (MATα, ura3-52, his3-200, ade2-101, trp1-901, leu2-3, gal4Δ, gal80Δ, met−, URA3::GAL1UAS-GAL1TATA-lacZ MEL1; prey construct) and transformants were selected on synthetic defined medium lacking tryptophan (SD-W). The pGADT7–Rep15 constructs were transformed in the yeast strain L40ΔGal4 (MATa, ura3-52, his3-200, ade2-101, trp1-901, leu2-3, gal4Δ, gal80Δ, URA3::opLEXA-LacZ, LYS2:opLEXA-HIS3; bait construct) and transformants were selected on synthetic defined medium lacking leucine (SD-L). For the mating experiments, 50 µL of bait and prey culture were added to 100 µL YPDA (Yeast peptone dextrose adenine) media in a 96-well plate (Sarstedt) and incubated 20-22 h at 30 °C with 180 r.p.m. Cells were resuspended and 5 µL were spotted on SD-LW (lacking leucine and tryptophan) plates as mating control, SD-LWH (lacking leucine, tryptophan and histidine) and SD-LWH + 2.5 mM 3-Aminotriazol (3′-AT, Sigma Aldrich), an inhibitor of the histidine biosynthesis. Plates were incubated at 30 °C for 3–4 days and the strength of interaction was estimated from the comparative growth of yeast cells on SD-LWH or SD-LWH + 3′-AT plates.

**Fluorescence microscopy**. The full-length human mCherry-tagged active Rab constructs (Rab3A$_{Q81L}$, Rab3B$_{Q81L}$, Rab3C$_{Q89L}$ and Rab3D$_{Q81L}$) were cloned into the pmCherry (C1) vector between XhoI and BamHI sites by conventional PCR using Y2H active Rab constructs as a template. The Rep15 was cloned into the pEGFP (N1) vector between XhoI and SalI sites and all constructs were verified by DNA sequencing. A detailed overview of constructs is presented in Supplementary Table 3. For transfection, Cos7 cells (ATCC: CRL-1651) were maintained in DMEM medium supplemented with 10% fetal bovine serum, 2 mM L-glutamine, and penicillin/streptomycin at 37 °C in the presence of 5% CO$_2$. Cells were grown on a coverslip in 6 well plates until they reached 60–70% confluency and transiently transfected using linear polyethylenimine, MW 25000 (PEI, Polysciences Inc, 3:1 PEI:DNA (12:4 µg). Expression was checked 16-24 h post transfection and cells were fixed with 3.7% paraformaldehyde in PBS for 15 min at room

temperature. After washing with PBS, coverslips were mounted on glass slides with SlowFade Gold antifade reagent (Invitrogen). Images were taken by an EVOS fl fluorescence microscope equipped with 60X/1.42 Plan Apochromat oil immersion objective. Images were assembled with Adobe Illustrator.

**Recombinant protein expression and purification.** Rab GTPases were expressed and purified as described previously[15]. Briefly, 6His-tagged Rab15, 6His-tagged Rab3 paralogs, 6His-tagged MBP-Rab34$_{Q111L\_1-237}$ were expressed into E. coli BL21 Codon-Plus (DE3)-RIL (Agilent Technologies) in lysogeny broth (LB) media supplemented with proper antibiotics and cells were lysed into a buffer containing 50 mM Hepes pH 8.0, 500 mM NaCl/LiCl, 1 mM MgCl$_2$, 10 μM GDP and 2 mM βME (2-Mercaptoethanol), 1 mM PMSF (Phenylmethylsulfonyl fluoride) and subsequently disrupted by passing through a fluidizer (Microfluidic). Lysates were cleared by centrifugation at 75,600 g for 30 min and the supernatant was loaded onto Ni Sepharose (HiTrap, GE Healthcare) column. The bound proteins were eluted with an imidazole gradient (lysis buffer supplemented with 500 mM Imidazole pH 8.0). The His6-tag/His6-tag-MBP was cleaved by Tobacco Etch Virus (TEV)-protease during overnight dialysis at 4 °C in a buffer (20 mM Hepes pH 8.0, 100 mM NaCl, 1 mM MgCl$_2$, 10 μM GDP and 2 mM βME) and a second Ni$^{2+}$-affinity purification was performed to remove the TEV protease and His6-tag/His6-tag-MBP. Rabs were preparatively loaded with GppNHp (Guanosine-5′-[β-γ-Imido]-triphosphate) or mant-GppNHp (2′/3′-O-N-Methyl-anthraniloyl)-guanosine-5′-[β-γ-imido]-triphosphate) or mant-dGppNHp (3′-O-(N-Methyl-anthraniloyl)−2′-deoxyguanosine-5′-[β-γ-imido]-triphosphate) and the reaction was performed, as described previously[15]. Further purity was achieved by gel filtration (HiLoad Superdex75 26/60, GE Healthcare) (Buffer: 20 mM Hepes 8.0, 50 mM NaCl, 1 mM MgCl$_2$ and 2 mM DTE). Residual MBP is removed by the amylose column.

Rab34 is loaded with GTP, the reaction was performed in presence of 10x EDTA and 50x GTP and reaction was performed at RT for 1 hr and a further 2 hr on ice, and buffer was exchanged on Amicon® Ultra-15 10 K NMWL centrifugal filter devices (Millipore). Nucleotide exchange efficiency was quantified by C18 reversed-phase column (Prontosil C18, Bischhoff Chromatography) with HPLC in 50 mM potassium phosphate buffer pH 6.6, 10 mM tetrabutylammonium bromide and 12% acetonitrile (v/v) and for the mant-GppNHp/mant-dGppNHp exchange the running buffer contains 25% acetonitrile (v/v). Protein samples were heat precipitated at 95 °C for 5 min and centrifuged at 15700 g for 10 min and loaded (25 μM, 20 μl) on the column. Peaks were integrated and to determine the nucleotide retention times; a nucleotide standard run was performed.

Wild type and mutant His6-tagged MBP-Rep15 fusion proteins were recombinantly expressed in E. coli BL21 Codon-Plus(DE3)-RIL cells in LB media supplemented with proper antibiotics and cells were grown at 37 °C until the optical density at 600 nm reached 0.8–1.0. Cells were stored at 4 °C for 20 min then 0.5 mM IPTG was added to induce expression and cells were allowed to grow at 20 °C for 14–16 h. Cells were pelleted down and stored at −80 °C until ready for purification. Cells were mechanically lysed by passing through a fluidizer (Microfluidic) in a buffer [Buffer A: 50 mM Tris pH 8.0, 500 mM NaCl, 2 mM βME, 1 mM PMSF and 1 mM CHAPS and lysates were cleared by centrifugation at 75,600 g at 4 °C for 30 min using a JA 25.50 rotor (Beckman-Coulter). Next, the proteins were purified by Ni$^{2+}$-affinity chromatography and protein fractions were eluted with an imidazole gradient (Buffer B: buffer A supplemented with 500 mM imidazole pH 8.0). The His6-tag MBP was cleaved by Tobacco Etch Virus (TEV)-protease during overnight dialysis at 4 °C in a buffer (50 mM Tris pH 8.0, 100 mM NaCl and 2 mM βME) and a second Ni$^{2+}$-affinity purification was performed to remove the TEV protease and His6-tag-MBP. Further purity was achieved by gel filtration (HiLoad Superdex75 26/60, GE Healthcare) (Buffer: 20 mM Tris 8.0, 100 mM NaCl and 2 mM DTE). Depending on purity, the protein was further purified by anion exchange chromatography (POROS™ 50 HQ strong anion exchange resin; Thermo Scientific™). The purified protein was collected and concentrated and washed with final buffer (Storage buffer: 20 mM Tris pH 8.0, 100 mM NaCl and 2 mM DTE) using Amicon® Ultra-15 10 K NMWL centrifugal filter devices; flash-frozen in liquid N$_2$ and stored at −80 °C. Protein concentration was determined using Bradford assay (Bio-Rad).

For the Rep15 ΔN1:Rab15$_{Q67L\_1-176}$, Rep15 ΔN1:Rab3A$_{Q81L\_\Delta CaaX}$ and Rep15 ΔN1: Rab3B$_{Q81L\_18-190}$ complex co-purifications, Rabs (untagged) were cloned into pET30 vector between NdeI and XhoI restriction sites. His6 tagged-Rep15 and Rab were co-expressed in E. coli BL21 D3-RIL cells and cells were lysed in a buffer containing 50 mM Hepes pH 8.0, 150 mM NaCl, 1 mM MgCl$_2$, 2 mM βME, 1 mM PMSF without CHAPS. The complexes were purified by Ni$^{2+}$-affinity chromatography as described above. Final purity was achieved by gel filtration in the final storage buffer (20 mM Hepes pH 8.0, 50 mM NaCl, 1 mM MgCl$_2$ and 2 mM DTE).

**Analytical size-exclusion chromatography.** The Rep15: Rab complex formation was analyzed by analytical size exclusion chromatography (aSEC). 110 μM of Rep15 and 121 μM of Rab protein (Effector: Rab stoichiometry of 1: 1.1) were mixed in a buffer containing 20 mM Hepes 8.0, 50 mM NaCl, 1 mM MgCl$_2$ and 2 mM DTE (Rab buffer). Samples were centrifuged for 15 min at 15700 g at 4 °C and 40 μl of the sample was injected into a Superdex 75 10/300 GL gel filtration

column (GE Healthcare) pre-equilibrated with the buffer with a flow rate of 0.5 ml/min at room temperature and absorption at 280 nm was recorded.

**Transient kinetic measurements.** Rep15: mant-GppNHp Rab15 kinetic measurements were performed in Rab buffer (20 mM Hepes 8.0, 50 mM NaCl, 2 mM MgCl$_2$ and 2 mM DTE) using a SX-20 stopped-flow apparatus (Applied Photophysics) at 25 °C. The experiments were performed using the signal from the methylanthraniloyl group of mant-GppNHp/mant-dGppNHp, excited with 365 nm and emission was detected through a 420 nm cutoff filter. Change in the fluorescence polarization of mant signal of 1 μM Rab15$_{mant-GppNHp}$ (post-mix) was observed with an increasing amount of Rep15. All stopped-flow results that were analyzed are averages of 6–8 individual traces. Single/double exponential functions were fit using the Origin9 software (OriginLab).

For the Rep15: mant-GppNHp kinetics measurements, a change in the fluorescence intensity of mant signal of 0.5 μM Rab3A$_{mant-GppNHp}$ (post-mix) was observed with an increasing amount of Rep15 (Buffer: 20 mM Hepes 8.0, 50 mM NaCl, 1 mM MgCl$_2$ and 2 mM DTE).

**Isothermal titration calorimetry.** Rep15-Rab interaction measurements were conducted by isothermal titration calorimetry (ITC) using an ITC200 or PEAQ-ITC microcalorimeter (MicroCal). A typical ITC experiment is consisted of 19 injections, with the first injection of 0.5 μL followed by 18 injections of 2 μL of Rab in a buffer containing 20 mM Hepes 8.0, 50 mM NaCl, 1 mM MgCl$_2$ and 1 mM Tris (2-carboxymethyl) phosphine (TCEP). To remove buffer mismatch, proteins were dialyzed overnight in the buffer and samples were centrifuged at 15700 g for 30 min at 4 °C. Protein concentration was determined by the Bradford assay (Bio-Rad). 500/600 μM of GppNHp/GTP Rab was titrated into the cell containing 50/60 μM Rep15. For the control experiments, the buffer was titrated into the cell containing the Rep15 and in the second control experiment, the Rab was titrated against the buffer. The binding isotherms were integrated, corrected for the offset and the data were fitted to a one-site-binding model using Origin 7.0 (MicroCal). The reported ITC results are representative of at least three independent measurements. Thermodynamic parameters are summarized in Supplementary Table 4.

**Crystallization, data collection and structure determination.** For all protein complexes, initial crystallization condition screens were performed with the JCSG Core I–IV, Pact, and Protein Complex suites (Qiagen). The sitting-drop vapor diffusion method was used, with a reservoir volume of 70 μl and a drop volume of 0.1 μl protein (300 μM complexes, 1:1 Rab:effector) and 0.1 μl reservoir solution at 20 °C. The best conditions were then optimized using the sitting-drop vapor diffusion method varying drop sizes to obtain well diffracting crystals. The complex of Rep15 (human): Rab3C$_{GppNHp\_Q222H\_10-227}$ (bovine) (150 μM of 1:1 complex) was crystallized in 0.1 M MES pH 6.0 and 10% (w/v) PEG 6000. The complex of human Rep15:Rab3B$_{GppNHp}$ (300 μM of 1:1 complex) was crystallized in 0.1 M Bis-Tris pH 7.5, 0.2 M NaI and 20% (w/v) PEG 3350. Further, we crystallized co-purified human Rep15 ΔN1:Rab3B$_{Q81L\_18-190}$ (22 mg/ml) in 0.1 M Bis-Tris pH 8.5, 0.2 M KNa-tartrate and 20% (w/v) PEG 3350. The crystals were briefly soaked in a reservoir solution supplemented with 10% glycerol and flash-cooled in liquid nitrogen. Diffraction data were collected at 100 K on beamline X10SA at the Swiss Light Source (Paul Scherrer Institute, Villigen, Switzerland). A native data set for the Rep15: Rab3C$_{GppNHp\_Q222H\_10-227}$ (bovine) complex crystal was collected at a wavelength of 0.977930 Å whereas a data set from the Rep15:Rab3B$_{GppNHp}$ complex at a wavelength of 0.999961 Å. A native data set was collected for the Rep15Δ N1:Rab3B$_{Q81L\_18-190}$ complex at a wavelength of 0.977930 Å. Data were integrated and scaled with XDS[62].

The Crystal of Rep15:Rab3C$_{GppNHp\_Q222H\_10-227}$ (bovine) complex diffracted to a resolution of 2.52 Å (space group I222 with a = 58.29 Å, b = 108.795 Å, c = 172.027 Å, α = 90.0˚, β = 90.0˚ and γ = 90.0˚) and a single copy of complex is present in the asymmetric unit of the crystal. The initial model for Rep15:Rab 3C$_{GppNHp\_Q222H\_10-227}$ (Bovine) complex was obtained by molecular replacement using PHASER[63] with the structure of the Rab3A (PDB: 3RAB)[35] as a search model. The partial model was completed with PHENIX AutoBuild[36] and manual building in Coot[64]. The final models were refined to convergence with phenix.refine[65]. For the Rep15:Rab3B$_{GppNHp}$ complex, the crystal diffracted to a resolution of 2.75 Å (space group C2 with a = 188.81 Å, b = 59.22 Å, c = 109.02 Å, α = 90.0˚, β = 111.90˚ and γ = 90.0˚) and for the Rep15ΔN1:Rab 3B$_{Q81L\_18-190}$ complex, the crystal diffracted to a resolution of 2.8 Å (space group P2$_1$ with a = 59.84 Å, b = 107.97 Å, c = 91.10 Å, α = 90.0˚, β = 106.467˚ and γ = 90.0˚) and two copies of the complex constitute the asymmetric unit. Rep15:Rab3C$_{GppNHp\_Q222H\_10-227}$ (bovine) complex was used as a model for molecular replacement.

Data collection and refinement statistics are summarized in Supplementary Table 1. Structural figures were prepared using PyMOL (DeLano Scientific; http://www.pymol.org/).

**Cell culture, generation of Rep15 knockout (KO) and validation.** U138MG (ATCC: HTB-16™) glioblastoma cell line was chosen as they show the maximum level of RNA as per Human Protein Atlas database. Of note, many different cell

lines were checked for the expression of the Rep15 by real-time quantitative PCR analysis, they all showed very low levels of expression. U138MG cells showed slightly better expression among the tested cell lines, although the expression was very low in these cells also. Further, the cells were cultured in DMEM medium supplemented with 10% FBS, L-Glutamate, Na-Pyruvate and penicillin-streptavidin antibiotic. Two guide RNA (sg1 5′—ACTGACCACCTCACAGACGA −3′ and sg2 5′—TTAATAGGAGAGGATTGCCT 3′) were chosen to delete the majority of the coding region of the Rep15 gene and were cloned into lentiCRISPR V2 vector (Addgene; Plasmid # 52961, a gift from Dr. Feng Zhang). The cells were transduced and selected with puromycin for a week before the single-cell dilutions were done in 96 well plates to generate single-cell clones. The genomic DNA was isolated from single-cell clones and PCR was done using primer pairs (FP 5′ - GAACCATTAACAGATGTGGCCT −3′ and RP 5′ - TGGAGGATATTCAA ATCCAAGG −3′) flanking the two guide RNA to confirm the deletion of the coding region (Supplementary Table 3).

For the generation of the rescue cell lines, cells were transduced with lentiviral vectors and selected with antibiotic (Hygromycin B) for 3–4 weeks before the experiments were performed. As strong overexpression of Rep15 also led to inhibition of Rep15 mediated function such as receptor mediated recycling of the transferrin, we waited for the expression to decrease slightly in our mixed stable cell population before performing the rescue experiments. The overexpression of the WT and Q107A mutant was validated by western blot against the Flag-tag and tublin served as a loading control (Fig. 9b, e).

**Reverse transcription quantitative PCR (qRT-PCR)**. RNA was extracted using NucleoSpin RNA, Mini kit for RNA purification (MACHEREY-NAGEL Inc) and cDNA was synthesized using Verso cDNA Synthesis Kit (ThermoFisher Scientific), as per manufactures instructions. Real-time PCR amplification and analysis were performed using a LightCycler 480 with DyNAmo ColorFlash SYBR Green (Roche, Mannheim, Germany) and the primers used in the study are listed in Supplementary Table 4. Calculations were done using the $\Delta\Delta$ cycle threshold (Ct) method[66]. For statistical analysis, the Student's t-test (two-tailed) was applied.

**Cell Proliferation assay**. For cell proliferation measurements, 50,000 cells (UC, KO, KO$^{WT}$ and KO$^{Q107A}$) were seeded in a six-well plate format and they were trypsinized and counted on subsequent days (every day starting from day 2 till day 6). The results are from four independent biological repeats over four different passages.

**Wound healing assay**. For wound healing assay 50,000 cells were seeded in each well of 2 well silicone insert (ibidi Cat no. 80209), in a 12-well plate. After the overnight incubation, the inserts were carefully removed and the cells were washed with PBS before exchanging the medium containing 2% FBS. A picture of the gap was taken immediately which served as a 0 h time point. Subsequently, a picture was taken at every 24 h time point and a gap between the two-cell fronts was taken as an estimate for the cell migration. The experiment was repeated six independent times with different passages.

**Anchorage-independent cell growth**. For anchorage, independent assay 10,000 cells were used per 35 mm dish. The bottom layer was made by pipetting 3 ml of 0.5% noble agar (DIFCO, 0142-01; 0.5 ml of 3% agar in water mixed with 2.5 ml of 20% FBS containing medium), while the top layer consists of 0.3% agar (0.33 ml of 3% of agar in water mixed with 2.67 ml of 20% FBS containing medium) with the required number of cells. After pipetting the cell suspension, the agar was left to solidify, which had a jelly-like consistency. To prevent further drying of the agar and cells it was made sure that the cell culture incubator was properly humidified and each petri dish was layered with a fresh 500 µl medium per week. A well-formed colony was visible in one month, which was stained by pipetting 200 µl of 5 mg/ml of iodonitrotetrazolium chloride solution. The number of the colonies in each dish was counted using open available software open CFU (http://opencfu.sourceforge.net). The experiment was repeated six independent times with different passages.

**Transferrin internalization and recycling**. The efficiency of receptor-mediated endocytosis in wild-type (WT), knockout (KO) and the rescue cells were examined by following the uptake of Alexa-488-conjugated human transferrin (Thermo-Fisher Scientific). The cells were seeded into 12 well plates with coverslips and incubated in a complete medium overnight. The next day, the cells were washed with serum-free medium three times and incubated with serum-free medium for 2 h. The medium was replaced with a fresh serum-free medium containing Alexa-488-conjugated human transferrin (20 µg/mL). For transferrin recycling, cells were exposed to transferrin for 30 min, after washing with a fresh serum-free medium three times, the culture was fed with a fresh medium for transferrin chase. Cells were fixed (3.7% PFA in PBS) at 2 h post washing and pictures were taken by a ZEISS Axio Observer fluorescence microscope. The intensity of the internalized transferrin was measured by ImageJ software and significance was calculated using the Student's t-test (two-tailed). For the rescue experiment with transiently transfected GFP-tagged Rep15 constructs, Alexa-647-conjugated human transferrin (ThermoFisher Scientific) was used. The cells were seeded on the glass coverslip

and the transferrin uptake and recycling experiment were performed exactly as described above after 48 h post- transfection. The experiment was repeated three times independently.

**Bioinformatics**. Clustal Omega[67] was used to generate multiple sequence alignments. The protein interaction interfaces from the asymmetric unit were examined in detail using the PDBePISA server (Proteins, Interfaces, Structures and Assemblies)[40]. DALI server was used for structural comparison[37].

**Reporting summary**. Further information on research design is available in the Nature Research Reporting Summary linked to this article.

## Data availability

The plasmids and cell lines created in this study are available from the corresponding author upon request. The coordinates and structure factors generated in this study have been deposited in the Protein Data Bank (https://www.rcsb.org) and are available under the accession codes: PDB 8A4A [https://doi.org/10.2210/pdb8A4A/pdb] (Rep15:Rab3C$_{Q222H\_10-227}$). PDB 8A4C [https://doi.org/10.2210/pdb8A4C/pdb] (Rep15:Rab3B). PDB 8A4B [https://doi.org/10.2210/pdb8A4B/pdb] (Rep15 ΔN1: Rab3B$_{Q81L\_18-190}$). The previously published structures used in this study are available as PDB accession codes: 3RAB [https://doi.org/10.2210/pdb3RAB/pdb], 2EO0 [https://doi.org/10.2210/pdb2EO0/pdb], 1ZBD [https://doi.org/10.2210/pdb1ZBD/pdb], 4TKD [https://doi.org/10.2210/pdb4TKD/pdb], 2ZET [https://doi.org/10.2210/pdb2ZET/pdb], 3BC1 [https://doi.org/10.2210/pdb3BC1/pdb], 1YHN [https://doi.org/10.2210/pdb1YHN/pdb], 5SZI [https://doi.org/10.2210/pdb5SZI/pdb], 5LPN [https://doi.org/10.2210/pdb5LPN/pdb], 6ZSI [https://doi.org/10.2210/pdb6ZSI/pdb], respectively. AlphaFold structural prediction of Rep15 is available from the AlphaFold Protein Structure Database at https://alphafold.ebi.ac.uk/entry/Q6BDI9. The sequences used in this study were obtained from Uniprot and the accession codes are described in Source Data file. Other source data are also provided in Source Data file. Source data are provided with this paper.

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

## Acknowledgements
We thank the Swiss Light Source (SLS) X10SA beamline staff at the Paul Scherrer Institute, Villigen, Switzerland, for assistance during data collection. We thank the X-ray community at the Max Planck Institute Dortmund and Dr. Raphael Gasper-Schönenbrücher for help with data collection. We are thankful to Dr. Matthias Müller for help in fluorescence polarization experiments. We would like to thank Prof. Dr. Bruno Goud, Dr. Mandy Hannemann and Prof. Dr. Aymelt Itzen for providing the yeast Rab library and also for sharing the yeast transformation and mating protocols. We acknowledge financial support from the Max-Planck-Society and the Deutsche Forschungsgemeinschaft (grant GO 284/10-1 to R.S.G).

## Author contributions
A.R. and R.S.G. conceived and designed the study. A.R. and N.B. carried out protein expression/purification. A.R. carried out all biochemical studies, yeast-two hybrid assay, in vivo co-localization experiments and crystallization work. A.R. and I.R.V. determined the Xray-structures. AKS performed Rep15 knockout and functional experiments in U138MG cells. A.R., A.K.S, G.P., I.R.V. and R.S.G. analyzed and interpreted the data. A.R., A.K.S, R.S.G. wrote the manuscript with critical input from G.P. and I.R.V.

## Funding

## Competing interests
The authors declare no competing interests.
