## [Peer Review File · Nature Communications]

REVIEWER COMMENTS

Reviewer #1 (Remarks to the Author):

This study from Rai et al. investigates the interaction between Rep15 and multiple Rab GTPases, including Rab15, Rab3, and Rab34.

The authors identify Rab3 paralogs and Rab34 as new interactors of Rep15, which was previously found to be a Rab15 effector.

They use a Y2H screening approach to identify additional Rab interactors of Rep15, and they discover that Rab3A, Rab3B, Rab3D, and Rab34 interact with Rep15.

They determine using both ITC and stopped-flow approaches that the K_d for the Rep15:Rab15 and Rep15:Rab3 interactions are $\sim 0.5 \mu\text{M}$.

Using cell culture, they determine that the “GTP-locked” mutants of the Rab3 paralogs all colocalize well with Rep15.

The authors then determine the crystal structures of Rep15:Rab3B and Rep15:Rab3C complexes. The structure reveals that Rep15 adopts a fold that is different from any known Rab effectors. The closest structural homolog is a protein involved in resolving holiday junctions (DNA recombination intermediates). They then provide an extensive comparison to other available Rab:effector structures.

They then validate and analyze the observed Rep15:Rab3 interfaces by testing the impact of substitution mutations in residues at the interface.

They find that Rab27A and Rab8A, which are both closely related to Rab3, do not interact well with Rep15. To determine why Rep15 interacts well with Rab3 but not Rab27A or Rab8A, the authors compare sequences and then test their hypothesis. They find that a single position, which is F in Rab27A and Rab8A but Y in Rab3, can explain the specificity. Switching F to Y enables Rab27A to interact well with Rep15, and switching Y to F decreases the affinity of Rab3 for Rep15.

Finally, the authors determine that Rep15 depletion reduces cell proliferation, wound-healing, anchorage-independent growth, and transferrin receptor recycling.

Overall I found this to be a comprehensive analysis of Rep15 and its interactions with its Rab regulators. I have only minor suggestions for improvement:

1. The authors perform a DALI search of the PDB using their new Rep15 structure. It might be interesting for the authors to also use Rep15 in a DALI search against the alphafold human protein prediction database to see if there is similarity to any predicted structures of other proteins (this has recently become an option on the DALI server website).

2. on lines 227-228 the authors twice refer to beta “sheets” when I think they mean “strands” (i.e. the beta-sheet has 7 beta-strands)

Reviewer #2 (Remarks to the Author):

This manuscript by Rai et al. provides a biophysical and structural characterization of the Rab effector Rep15 and its interaction with cognate GTPases, and some functional data on Rep15. The key results include the identification of Rab3A/B/C/D and Rab34 as novel Rep15 effectors through a yeast two-hybrid screen. These interactions are validated by qualitative and quantitative interaction studies using ITC, analytical SEC and fluorescence polarization. A characterization of Rep15 binding to the known interactor Rab15 is described as well. The crystal structures of Rep15 bound to Rab3C and Rab3A provide insight into the molecular basis of binding, which is used for a mutational study to investigate the determinants of specificity in recognition. Finally, the initial characterization of a Rab15 knock-out cell line is presented.

This study adds a further piece to the understanding of Rab effectors. Several questions remain unanswered and the following aspects should be addressed.

1. While the investigation of Rep15 interaction with Rab3s and Rab15 is very thorough, Rab34 is mostly neglected. In Fig. 3, a quantitative determination of Rep5-Rab34 affinity by ITC would be informative.

Because Rab34 does not contain a Tyr in the RabF3 motif, which is important for Rab3 binding to Rep15, but a Phe, the authors suggest that Rep15 and Rab34 might interact via a different interface (l. 354). It would be interesting to see if a Phe to Tyr mutation increases Rab34 binding, which would argue for similar binding interfaces. Also, (some of) the Rep15 mutations shown in Figure 3 could be tested with Rab34 to establish if the same interface of Rep15 binds to Rab34 or not.

2. ITC titrations for Rab27A_F81Y (and wt as control) should also be performed for a quantitative comparison to Rab3A/B/C/D and Rab15 (and Rab 34).

3. The authors claim that Rep15 co-localizes with Rab3 to exocytic compartments. This is concluded from overexpression of both Rab3 and Rep15 in Cos7 cells. This only shows that both proteins can interact in cells, but are the visualized structures really secretory compartments? A respective endogenous marker should be included in the IF.

4. It may be an overstatement that “Y2H experiments clearly show that Rep15 does not interact with these [other] evolutionarily similar Rabs” (l. 291-292). Considering that Rab15 was a false negative in the Y2H screen, other Rabs might have been missed.

5. “This tyrosine residue is replaced by phenylalanine in other Rab8 family members (Rab8A/B, Rab10, Rab13 and Rab15)” (l. 300-302). Rab15 should not be listed here.

6. The Rep15 ko cells show slower growth, wound healing and anchorage independent proliferation, and impaired transferrin receptor recycling, which was also observed in Rep15 kd studies before. A quantification of the wound healing and colony formation assay should be provided.

Interestingly, expression levels of some interacting Rabs are downregulated. Are other Rabs (Rab8, Rab27) not affected?

The description of the ko cells is somewhat detached from the rest of the manuscript and does not add mechanistic insight. It might thus be interesting to show a reconstitution of the cells with wt Rep15 and a Rab-binding deficient mutant (e.g. Q107A) to establish a link between Rep15 Rab effector function and the cellular phenotypes.

7. The biophysical characterizations are often redundant. If ITC measurements are shown, analytical SEC data does not provide any additional information. k_{obs} , k_{on} and k_{off} are only determined for some interaction, which is fine. It may be helpful though to provide a table with the all determined constants listed for the different binding pairs.

Additional points:

- Some data in the main figures may be better suitable as supplement, e.g. Figs 1a, 1f, 8a-c

- Figure 5d and 6i apparently show the same data. One should be removed or replaced by a different repeat.

- How were the expression levels of Rabs in U138MG wt/Rep15 ko cells normalized (Supp Fig 10)?

The manuscript is in parts difficult to read and the writing seems to be not properly finished in some sections:

- “The mammalian Rab 3 subfamily is comprised of 4 paralogs (Rab3A, 3B, 3C, 94 and 3D) and physically.” (l. 95-96) Incomplete sentence

- “pretty similar” (l. 238)

- The meaning of l. 365-367 and connection to the paragraph is not clear.

- “listed in the supplementary table2” (l. 596) should refer to supp table 3

- “EGFP-Rep15 nicely co-localizing with mCherry tagged active Rab3 paralogs.” (l. 838-839)

- Legend for Figure 7 d-f is missing.

We thank the reviewers for their positive comments on our manuscript and constructive suggestions. In the revised manuscript, we have addressed all the reviewer's concerns by performing suggested additional experiments and by rewriting the manuscript.

Below is a point-by-point response (in blue) to all comments and a description of all the changes, we have made to our manuscript. We marked changed passages in the manuscript in red and refer to the relevant positions in the text in our comments here.

Reviewer #1 (Remarks to the Author):

This study from Rai et al. investigates the interaction between Rep15 and multiple Rab GTPases, including Rab15, Rab3, and Rab34.

The authors identify Rab3 paralogs and Rab34 as new interactors of Rep15, which was previously found to be a Rab15 effector.

They use a Y2H screening approach to identify additional Rab interactors of Rep15, and they discover that Rab3A, Rab3B, Rab3D, and Rab34 interact with Rep15.

They determine using both ITC and stopped-flow approaches that the K_d for the Rep15:Rab15 and Rep15:Rab3 interactions are ~0.5 μM.

Using cell culture, they determine that the "GTP-locked" mutants of the Rab3 paralogs all colocalize well with Rep15.

The authors then determine the crystal structures of Rep15:Rab3B and Rep15:Rab3C complexes. The structure reveals that Rep15 adopts a fold that is different from any known Rab effectors. The closest structural homolog is a protein involved in resolving holiday junctions (DNA recombination intermediates). They then provide an extensive comparison to other available Rab:effector structures.

They then validate and analyze the observed Rep15:Rab3 interfaces by testing the impact of substitution mutations in residues at the interface.

They find that Rab27A and Rab8A, which are both closely related to Rab3, do not interact well with Rep15. To determine why Rep15 interacts well with Rab3 but not Rab27A or Rab8A, the authors compare sequences and then test their hypothesis. They find that a single position, which is F in Rab27A and Rab8A but Y in Rab3, can explain the specificity. Switching F to Y enables Rab27A to interact well with Rep15, and switching Y to F decreases the affinity of Rab3 for Rep15.

Finally, the authors determine that Rep15 depletion reduces cell proliferation, wound-healing, anchorage-independent growth, and transferrin receptor recycling.

Overall I found this to be a comprehensive analysis of Rep15 and its interactions with its Rab regulators. I have only minor suggestions for improvement:

We thank the reviewer for the positive assessment of our manuscript.

1. The authors perform a DALI search of the PDB using their new Rep15 structure. It might be interesting for the authors to also use Rep15 in a DALI search against the alphafold human protein prediction database to see if there is similarity to any predicted structures of other proteins (this has recently become an option on the DALI server website).

We thank the reviewer for the suggestion; we have performed a DALI search against the AlphaFold database (AF-DB). However, we could not obtain more information from this search. The related proteins are diverse, and they encompass only half of the Rep15.

2. on lines 227-228 the authors twice refer to beta “sheets” when I think they mean “strands” (i.e. the beta-sheet has 7 beta-strands)

Thanks for the correction, changes have been made.

Reviewer #2 (Remarks to the Author):

This manuscript by Rai et al. provides a biophysical and structural characterization of the Rab effector Rep15 and its interaction with cognate GTPases, and some functional data on Rep15. The key results include the identification of Rab3A/B/C/D and Rab34 as novel Rep15 effectors through a yeast two-hybrid screen. These interactions are validated by qualitative and quantitative interaction studies using ITC, analytical SEC and fluorescence polarization. A characterization of Rep15 binding to the known interactor Rab15 is described as well. The crystal structures of Rep15 bound to Rab3C and Rab3A provide insight into the molecular basis of binding, which is used for a mutational study to investigate the determinants of specificity in recognition. Finally, the initial characterization of a Rab15 knock-out cell line is presented.

This study adds a further piece to the understanding of Rab effectors. Several questions remain unanswered and the following aspects should be addressed.

We thank the reviewer for the constructive suggestions and comments.

1. While the investigation of Rep15 interaction with Rab3s and Rab15 is very thorough, Rab34 is mostly neglected. In Fig. 3, a quantitative determination of Rep5-Rab34 affinity by ITC would be informative.

In the revised manuscript, the Rep15-Rab34 ITC data is included (Fig. 3j). Compared to Rab3s and Rab15, Rab34_{Q111L-237} shows the highest affinity of $\sim 0.1 \mu\text{M}$ ($64 \pm 32 \text{ nM}$) for Rep15 interaction and unlike Rab3s and Rab15, Rab34 interaction with the Rep15 is an entropy-driven process.

Because Rab34 does not contain a Tyr in the RabF3 motif, which is important for Rab3 binding to Rep15, but a Phe, the authors suggest that Rep15 and Rab34 might interact via a different interface (l. 354). It would be interesting to see if a Phe to Tyr mutation increases Rab34 binding, which would argue for similar binding interfaces.

Sorry for the confusion, we meant that apart from the switch regions (I/II), the N-terminal extension might contribute to the Rep15 binding interface and also the orientation of switch II residue could differ from Rab3. As per the reviewer's suggestion, Phe to Tyr mutation is introduced in Rab34_{Q111L-237} (F114Y of RabF3 motif). No change in the binding affinity was observed ($78.7 \pm 9.5 \text{ nM}$; Fig. 6k), suggesting binding interface of Rab34 is different from Rab3. Unlike Rab34_{Q111L-237}, both entropy and enthalpy-driven processes contribute to the free energy of the Rep15:Rab34_{Q111L-F114Y:237} complex.

Also, (some of) the Rep15 mutations shown in Figure 3 could be tested with Rab34 to establish if the same interface of Rep15 binds to Rab34 or not.

The Rep15-Rab34 interaction is an entropy-dependent process and enthalpy changes are small, resulting in a lower heat signal upon mixing Rab34^{wt} with Rep15 during ITC experiments. Due to the low signal, we opted for aSEC (qualitative) experiments with the Rep15 mutants (Supplementary Fig. 8) and the result is quite similar to Rab3A.

To have a complete picture, we have also performed Rep15 mutant aSEC experiments with Rab15 (Supplementary Fig. 9).

2. ITC titrations for Rab27A_F81Y (and wt as control) should also be performed for a quantitative comparison to Rab3A/B/C/D and Rab15 (and Rab 34).

Rab27A ($86.2 \pm 18.1 \mu\text{M}$, Fig. 6i) and Rab27A_F81Y ($10.8 \pm 1.8 \mu\text{M}$, Fig. 6j) ITC titration were performed. F81Y mutation leads to a more than 8-fold increase in the binding affinity.

3. The authors claim that Rep15 co-localizes with Rab3 to exocytic compartments. This is concluded from overexpression of both Rab3 and Rep15 in Cos7 cells. This only shows that both proteins can interact in cells, but are the visualized structures really secretory compartments? A respective endogenous marker should be included in the IF.

Rab3s are a marker of the secretory/exocytic compartment. However, in the context of this manuscript, our message is that the Rep15 interacts with Rab3s, suggesting that it might have a role in Rab3-dependent processes. So to avoid confusion, we have modified the text (p.6, 3rd paragraph).

4. It may be an overstatement that "Y2H experiments clearly show that Rep15 does not interact with these [other] evolutionarily similar Rabs" (l. 291-292). Considering that Rab15 was a false negative in the Y2H screen, other Rabs might have been missed.

We thank the reviewer for raising this important point. Yes, we did not see Rab15 in our Y2H experiments, suggesting the possibility of missing other potential Rab as a Rep15 interaction partner. We have added a cautionary note on this on p6 (paragraph 2) and p9 (last paragraph). Additional experiments are needed, for example, pull-downs against different human Rabs.

5. "This tyrosine residue is replaced by phenylalanine in other Rab8 family members (Rab8A/B, Rab10, Rab13 and Rab15)" (l. 300-302). Rab15 should not be listed here.

Thanks for the correction. The text is modified.

6. The Rep15 ko cells show slower growth, wound healing and anchorage independent proliferation, and impaired transferrin receptor recycling, which was also observed in Rep15 kd studies before. A quantification of the wound healing and colony formation assay should be provided.

As suggested by reviewers the quantification of the wound healing and colony formation assay has been included in the revised manuscript (Figure 8d and f).

Interestingly, expression levels of some interacting Rabs are downregulated. Are other Rabs (Rab8, Rab27) not affected?

We thank the reviewer for pointing this out. We have quantified the suggested Rabs (Rab8A and Rab27A), and the results are now included in the revised manuscript (Figure 8b). No changes in the expression level of these Rabs were observed in the KO as well as in the rescue cell lines.

The description of the ko cells is somewhat detached from the rest of the manuscript and does not add mechanistic insight. It might thus be interesting to show a reconstitution of the cells with wt Rep15 and a Rab-binding deficient mutant (e.g. Q107A) to establish a link between Rep15 Rab effector function and the cellular phenotypes.

Thank you for the suggestion. We have now included the rescue experiment both by wt Rep15 as well as a Rab-binding deficient mutant (Q107A). We could show that wt Rep15 could rescue the transferrin experiment, while Q107A mutant failed to do so (Figure 8j and l).

We further tested whether we see a similar rescue in other experiments. To our surprise, both the wt and Q107A mutant show similar mild rescue, indicating that these phenotypes might not be affected by Q107A mutation. Another important point to note is that the interpretation of the rescue experiments is made difficult in this instance, as overexpression of the Rep15 also strongly inhibits the endosomal recycling pathway as has been shown by Strick *et al.*

In addition, overexpression of the Rep15 led to some toxic effects on the cells, which we did not measure directly but was quite evident as we repeatedly failed to obtain single-cell clonal expansion in post-rescue experiments. A similar toxic effect on HeLa cells post overexpression of Rep15 constructs was observed by Strick *et al.* For our rescue experiments, we worked with a mixed population of the cells that were selected with antibiotics for over 3-4 weeks. By doing so, we had hoped to have a Rep15 expression that was not too high. The fact that strong overexpression led to the inhibition of the recycling pathway was further validated by the fact that upon transient transfection of knockout cells with GFP-tagged Rep15 did not show any rescue, while low Rep15-GFP expressing cells showed significant rescue. Further, we overexpressed Rep15 in the WT cells, which had an inhibitory effect on the recycling pathway (a similar observation was made by Strick *et al.* in HeLa cells). Due to the fact that both overexpression, as well as the knockout, have similar effects, this means that caution is appropriate in interpreting these experiments (p. 14, end of first paragraph).

7. The biophysical characterizations are often redundant. If ITC measurements are shown, analytical SEC data does not provide any additional information. $k(\text{obs})$, $k(\text{on})$ and $k(\text{off})$ are only determined for

some interaction, which is fine. It may be helpful though to provide a table with the all determined constants listed for the different binding pairs.

Supplementary Table 2 has been introduced for this purpose.

Additional points:

- Some data in the main figures may be better suitable as supplement, e.g. Figs 1a, 1f, 8a-c

Data is moved to supplementary figures (Supplementary fig 1a-b and supplementary figure 12).

- Figure 5d and 6i apparently show the same data. One should be removed or replaced by a different repeat.

Figure 5d is replaced with a different repeat and figure 6i is removed.

- How were the expression levels of Rabs in U138MG wt/Rep15 ko cells normalized (Supp Fig 10)?

Normalization of the quantitative real-time PCR of the Rabs was done by the $2^{(-\Delta\Delta Ct)}$ method². In brief, Ct values of target genes were normalized to the average Ct value of two housekeeping genes using the $\Delta\Delta Ct$ method and expressed as relative mRNA expression levels compared to the control group, which is set as 1.

The manuscript is in parts difficult to read and the writing seems to be not properly finished in some sections:

- “The mammalian Rab 3 subfamily is comprised of 4 paralogs (Rab3A, 3B, 3C, 94 and 3D) and physically.” (l. 95-96) Incomplete sentence

Sorry for the mistake. “and physically” is removed.

- “pretty similar” (l. 238)

Removed the word “pretty”. Rab3C is, expectedly, similar to Rab3A.

- The meaning of l. 365-367 and connection to the paragraph is not clear.

Lines 365-367 are moved to the previous paragraph. This paragraph describes other studied Rab3, Rab34 and Rab15 effectors.

- “listed in the supplementary table2” (l. 596) should refer to supp table 3

Modified.

- “EGFP-Rep15 nicely co-localizing with mCherry tagged active Rab3 paralogs.” (l. 838-839)

The text is modified. “Merged images show strong co-localization between EGFP-Rep15 and mCherry tagged active Rab3 paralogs”.

- Legend for Figure 7 d-f is missing.

Figure legend is included.

In addition to the changes suggested by the reviewers, we have some minor changes:

1: Rep15 mutants: Rab15 aSEC experimental data is included in the supplementary figure 9.

2: Some of the aSEC/ITC titrations were replaced with different repeats.

1. Strick, D.J. & Elferink, L.A. Rab15 effector protein: a novel protein for receptor recycling from the endocytic recycling compartment. *Mol Biol Cell* **16**, 5699-709 (2005).
2. Pfaffl, M.W. A new mathematical model for relative quantification in real-time RT-PCR. *Nucleic Acids Res* **29**, e45 (2001).

REVIEWERS' COMMENTS

Reviewer #2 (Remarks to the Author):

The revised version of this manuscript contains a substantial amount of additional data that complete this comprehensive and interesting study. The rescue experimnts with the Rep15 ko cell lines provide additional insight and show a more complicated picture, which will be interesting to explore in future studies.

All major and minor issues raised during initial review have been nicely addressed. Some new minor issues I noticed:

- l. 310: Figure number is missing
- significance markers fig 8a - day 6 is shifted

I have no further comments or objections.

REVIEWER COMMENTS

Reviewer #2 (Remarks to the Author):

The revised version of this manuscript contains a substantial amount of additional data that complete this comprehensive and interesting study. The rescue experimnts with the Rep15 ko cell lines provide additional insight and show a more complicated picture, which will be interesting to explore in future studies.

We thank the reviewer for the positive feedback.

All major and minor issues raised during initial review have been nicely addressed. Some new minor issues I noticed:

- l. 310: Figure number is missing

Figure number is included.

- significance markers fig 8a - day 6 is shifted

Thanks for the observation, we have corrected the position of the significance markers.

I have no further comments or objections.